# DAMO: Decoding by Accumulating Activations Momentum for Mitigating Hallucinations in Vision-Language Models

**Kaishen Wang**[†][*]**, Hengrui Gu**[‡][*]**, Meijun Gao**[ℓ][*]**, Kaixiong Zhou**[‡]
[†]University of Maryland, College Park [‡]North Carolina State University [ℓ]Michigan State University
{kaishenw7322,guhenry217}@gmail.com,gaomeiju@msu.edu,kzhou22@ncsu.edu

## Abstract

Large Vision-Language Models (LVLMs) exhibit significant potential in multimodal tasks but often struggle with hallucinations—responses that are plausible yet visually ungrounded. In this work, we investigate the layer-wise prediction tendencies of LVLMs and conduct an in-depth analysis of their decoding mechanism. We observe that LVLMs tend to "overthink" during the final stages of decoding, making significant prediction shifts in the last few layers often favoring incorrect results, which leads to a surge in hallucinative outputs. Leveraging this localized pattern, we propose a novel decoding strategy inspired by the momentum analogy used in gradient descent-based optimizers. Our method enforces decoding consistency across layers in an adaptive manner during forward passes—an under-explored approach in existing works. This strategy significantly improves the reliability and performance of LVLMs in various multimodal tasks, while introducing only negligible efficiency overhead. The code is available at https://github.com/tunantu/DAMO.

## 1 Introduction

The recent advancement in model architecture and training methodologies has led to unprecedented development and wide adoption of Large Vision-Language Models (LVLMs) (Devlin, 2018; Chen et al., 2019; Liu et al., 2024c; Zhou et al., 2023a; Ye et al., 2023b; Peng et al., 2025). Through bridging the gap between visual and textual modalities, they offer a viable and promising solution for various multimodal tasks such as visual question answering and image captioning (Liu et al., 2024d; Ye et al., 2023a; Zhu et al., 2023b; Li et al., 2023b; Lee et al., 2024). Despite their success, LVLMs continue to struggle with hallucinations (Liu et al., 2023a; Yin et al., 2023), a phenomenon where they tend to generate syntactically plausible yet visually ungrounded responses (see Fig. 1a). This intractable challenge significantly undermines users' trust in their output, thereby hindering their broader application in real-world scenarios (Chen et al., 2024b; Hu et al., 2023).

Several studies have delved into the mechanisms behind hallucinations, attributing them to factors such as over-reliance on statistical pre-training biases (Agarwal et al., 2020; Agrawal et al., 2016), language priors, and attention deficiency (An et al., 2024). All these proposed conjectures suggest that the inherent neglect of modality information integrated at later stages can lead to inaccurate outputs. Additionally, indiscriminate associative reasoning on linguistic and visual data can cause hallucinated responses to gradually dominate the decoding distribution. Thus they leverage knowledge editing methods (De Cao et al., 2021; Meng et al., 2022; Gu et al., 2024a; Khandelwal et al., 2024; Shi et al., 2024; Gu et al., 2024b; Zhong et al., 2024; Liu et al., 2023c; Jiang et al.) to inject new answers or knowledge into foundation models and to erase hallucination on mis-answered samples. The existing methods edit knowledge of LVLMs by either fine-tuning specific memory parameters or maintaining an external memory of updated knowledge facts. Despite these efforts, we propose a critical doubt remains unresolved: _Does LVLMs really don't know ground-truth answers for solving those hallucinated problems?_

---

[*]These authors contributed equally. The order of authorship is decided through dice rolling.

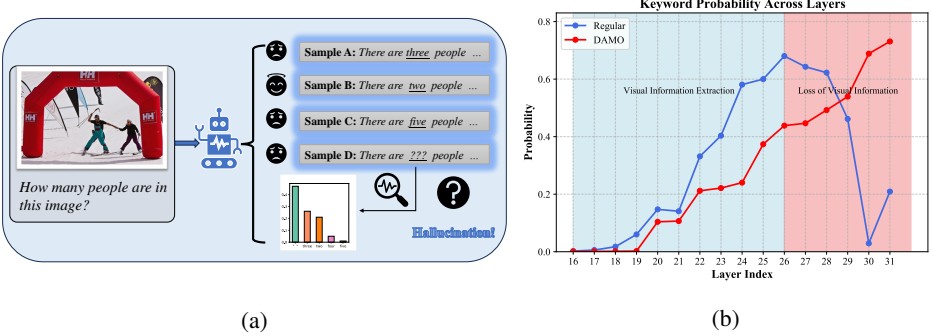

(a)                                                                 (b)

Figure 1: (a) Hallucination example: the model is asked "How many people are in this image?" for an image of two people skiing. And the analysis of probability distribution for key token. (b) Probability of predicting the token *'two'* across layers. The blue curve shows Regular decoding, while the red shows the effect of DAMO. The shaded area marks the stages of *Visual Information Extraction* and *Loss of Visual Information*.

This suspicion is originated from following observation that LVLMs generally can decode the desired outputs at early layers but later they turn to focus on other unrelated contents.

To answer this question, we analyze the layer-wise prediction tendencies of LVLMs and summarize the patterns of hallucination progressing accordingly. Specifically, we leverage a vanilla MiniGPT-4 Zhu et al. (2023c) and manually collect an evaluation dataset on visual question answering task from COCO val2014 Lin et al. (2014). Motivated by the idea of *early exit* (Teerapittayanon et al., 2016; Elbayad et al., 2019; Schuster et al., 2022), we decode the immature hidden states exited by LVLMs in advance and pay attention to confidence variations of the ground-truth token across layers. A qualitative illustration of this phenomenon is shown in Fig. 1b: The decoding probability of the ground-truth answer, '*two*', steadily increases across the early layers and then dominates the decoding distribution, revealing that LVLMs internally owns the capability to extract the fine-grained visual semantics. However, its decoding tendency drastically deviates at the later layers and token '*two*' is not the first choice anymore at the last layer. Based on this observation, we hypothesize that the hallucinations within LVLMs manifest as **localized surges** at the later layers, which in turn suppresses the pre-existing and visual related information in the decoding distribution, ultimately leading to visually ungrounded responses.

To address these concerns and improve the truthfulness of LVLMs, we propose a novel decoding method, **D**ecoding by **A**ccumulating Activations **MO**mentum (DAMO). Specifically, DAMO accumulates a momentum by aggregating the activations layer-by-layer and uses this momentum, which contains the historical updating trends, to correct hidden states at the later layers of LVLMs. This approach amplifies visual semantics consistently extracted throughout the inference while reducing hallucination biases that are intensively introduced at the later layers. This idea is built upon two foundations: ❶ In gradient descent, momentum has been widely proven to an effective acceleration technique by accumulating persistent updates towards low-curvature directions, driving the model closer to the global optimum (Sutskever et al., 2013; Polyak, 1964). ❷ Ahn et al., 2024 have demonstrated that a transformer can implement preconditioned gradient descent, and that the forward computation process of a $L$-layer transformer is conditionally equivalent to $L$ steps of gradient descent. Therefore, implementing "momentum" in the forward computation of transformers to counteract the localized and intense emergence of hallucinations is an intuitive and expectedly effective approach. Fig. 1b shows that DAMO successfully reduces the hallucinations that occur during the inference of the vanilla MiniGPT-4.

Experiments on MME and POPE datasets across various LVLMs demonstrate that DAMO significantly mitigates hallucinations, resulting in more visually grounded and accurate predictions. When applied to mPLUG-Owl2 Ye et al. (2023b) without any additional training, DAMO achieved a remarkable 100-point improvement over regular decoding on the MME dataset, outperforming all other baseline methods. Bulilding on this success we further extended DAMO to LLMs, where is also delivered strong results on benchmarks such as TruthfulQA Lin et al. (2021), FACTOR Muhlgay et al. (2023), and GSM8K Cobbe et al. (2021), indicating that DAMO possesses notable transferability across different tasks and model types. The contributions are summarized as follows:

- We identify that hallucinations in Large Vision-Language Models (LVLMs) primarily occur during the inference process and are driven by localized surges in the later layers, which suppress visual information.
- We introduce Decoding by Accumulating Activations Momentum (DAMO), a novel approach that reduces hallucinations by accumulating activation momentum, significantly improving visual grounding in LVLMs across multiple benchmarks.

## 2 RELATED WORK

**Large Vision-Language Models** The evolution of Large Vision-Language Models (LVLMs) has progressed from BERT-based models (Devlin, 2018; Lu et al., 2019; Tan & Bansal, 2019; Chen et al., 2019) to those integrated with Large Language Models (LLMs) (Liu et al., 2023b; Zhu et al., 2023c; Zhou et al., 2023a; Ye et al., 2023b), which have significantly improved their capabilities. Early models like ViLBERT Lu et al. (2019), LXMERT Tan & Bansal (2019), and UNITER Chen et al. (2019) effectively merged visual and textual features using BERT-style architectures. The introduction of LLMs enabled LVLMs such as CLIP Radford et al. (2021) and ALIGN Jia et al. (2021), which significantly enhanced adaptability and performance through end-to-end training. Recent works like LLaVA (Liu et al., 2023b; 2024b) and InstructBLIP Dai et al. (2023) further refined these models using visual instruction fine-tuning, demonstrating adaptability across diverse vision-language tasks and showcasing a growing trend toward task-specific approaches.

**Hallucinations in LVLMs** In Large Language Models (LLMs), hallucinations have been extensively studied (Ji et al., 2023; Yao et al., 2023; Zhang et al., 2023b;a; Liu et al., 2024a; Xu et al., 2024), particularly in contexts where generated text diverges from the input or factual reality. In LVLMs, hallucinations present additional challenges due to the alignment between visual and textual data (Biten et al., 2022; Li et al., 2023c; Wang et al., 2024), which increases the potential for hallucinations. This issue is particularly prevalent in tasks such as image captioning and visual question answering (Liu et al., 2023a; Yin et al., 2023; Zhou et al., 2023b; Zhu et al., 2024a; Gunjal et al., 2024), where maintaining coherence between modalities is critical.

**Addressing Hallucinations in LVLMs** Some researches (Gunjal et al., 2024; Wang et al., 2024) have attempted to mitigate hallucinations through fine-tune models. However, this approach can be resource-sensitive, leading to increased computational costs, while also risking a decline in performance on other tasks and limiting the model's overall versatility. Additionally, there are also some approaches (Zeng et al., 2021; Kim et al., 2023; Zhou et al., 2023b) trying to enhance alignment between visual and text modalities to address hallucinations.

At the same time, DoLA Chuang et al. (2023) has been proposed to address hallucination issues in LLMs through contrasting decoding, achieving promising results without the need for additional training. Subsequently, contrasting decoding was applied in LVLMs to address hallucination issues. VCD (Visual Contrastive Decoding) Leng et al. (2024) is a training-free method designed to mitigate object hallucinations in LVLMs, contrasting output distributions from original and distorted visual inputs to reduce reliance on statistical bias and unimodal priors. HIO (Hallucination-Induced Optimization) Chen et al. (2024a) addresses hallucinations in LVLMs by amplifying the contrast between hallucinatory and targeted tokens using a fine-tuned Contrary Bradley-Terry Model. IBD (Image-Biased Decoding) Zhu et al. (2024b) mitigates hallucinations in LVLMs by contrasting predictions from a standard LVLM with those from an image-biased LVLM, amplifying image-related information and reducing over-reliance on text. The common issue with these methods is that they only correct the next token prediction through logits contrasting at the final layer, without attempting to identify and address hallucinations in the hidden states during the inference process. In addition, OPERA Huang et al. (2024) mitigates hallucination by reducing reliance on summary tokens during decoding and adjusting token selection based on previously generated tokens.

## 3 METHOD

We first introduce the decoding process in LVLMs to provide some preliminaries. Next, we elaborate our motivation to transfer the idea of momentum into LVLM decoding by an intuitive example. Finally, we propose a novel, seamlessly integrated decoding method called DAMO, which accu-

mulates the raw activation updates into the momentum layer by layer to correct the hidden states' updating direction, effectively smoothing the update process and mitigating hallucinations.

## 3.1 DECODING IN VISION-LANGUAGE MODELS

Given an input representation $h_t^0$ at time step $t$, the standard decoding process of a $N$-layer LVLM is outlined as follows (Liu et al., 2024c; Zhu et al., 2023a; Li et al., 2023a):

$$\begin{cases} h_t^{j+1} = f_j(h_t^j) + h_t^j, & j = 0, \ldots, N-1, \\ p(x_{t+1}|x_{:t}) = \mathrm{softmax}(\phi(h_t^N)), \end{cases} \tag{1}$$

where $f_j(\cdot)$ refers to the $j$-th transformer layer, and $h_t^j$ denotes the hidden states at layer $j$ at time step $t$. $f_j(h_t^j)$ represents the activation output by layer $j$, which is the sum of the activations produced by both the MLP and the self-attention modules at the current layer. $\phi(\cdot)$ represents the language head that predicts the next-token probability over the entire vocabulary. Although $\phi(\cdot)$ is only trained to transform the final-layer activation (i.e., $h_t^N$) into the next-token distribution, several studies (Teerapittayanon et al., 2016; Elbayad et al., 2019; Schuster et al., 2022; Zhou et al., 2025; Ouyang et al., 2025) have demonstrated that applying the language head directly to intermediate activations exited from earlier layers can also produce meaningful distributions, reflecting prediction tendencies of LVLMs. We formally describe *early exit* as follows:

$$p_j(x_{t+1}|x_{:t}) = \mathrm{softmax}(\phi(h_t^j)), \quad j = 1, \ldots, N-1. \tag{2}$$

Here, $h_t^j$ refers to the intermediate activation exited by layer $j$ at time step $t$. In Section 3.2, we utilize this technique to illustrate our motivation.

## 3.2 MOTIVATION

In Visual Question Answering (VQA), LVLMs take an image and a related question as input, producing an answer based on visual cues. We conducted a preliminary experiment on a small-scale, self-constructed dataset to explore hallucinations in LVLMs. The dataset, derived from randomly selected images from COCO val2014 Lin et al. (2014), included questions focused on quantifiable attributes such as color, number, and direction. We identified 100 instances exhibiting hallucinations and analyzed the probability distribution of key words (e.g., 'two' in Figure 1b) across the model's layers during inference.

Our findings revealed that 75% of the samples shared a consistent issue: while LVLMs are adept at extracting visual information from images, hallucinations frequently emerge in the later layers of the inference process. This analysis highlighted two key insights: ❶ LVLMs are already proficient in capturing detailed visual information, so further intensifying image-text fusion is unnecessary. ❷ Hallucinations primarily occur during later inference stages, so we only need to correct hallucinations at these stages of the inference process.

Motivated by this, we propose DAMO that refines activations by leveraging earlier layer information, helping to preserve visual context and reduce hallucinations, ultimately improving prediction accuracy.

## 3.3 DECODING BY ACCUMULATING ACTIVATION MOMENTUM

**Momentum in Gradient-based Optimization** Momentum, a popular technique in gradient-based optimization, has been proven to be fairly effective for providing better convergence rate (Polyak, 1964). The idea of this method is to preserve the historical updating trends from previous steps in the momentum and use this value to correct the update direction of the current step. Specifically, the model parameters $\theta_t$ at the current step $t$ are obtained according to the following formula:

$$\begin{cases} v_t = \beta v_{t-1} + (1-\beta)\nabla L(\theta_{t-1}), \\ \theta_t = \theta_{t-1} - \eta v_t. \end{cases} \tag{3}$$

| Starting Layer | Existence | Count | Position | Color | Posters | Celebrity | Scene | Landmark | Artwork | OCR | Total |
|---|---|---|---|---|---|---|---|---|---|---|---|
| Vanilla Decoding | 190.00 | **160.00** | 138.33 | 165.00 | 141.50 | 135.88 | 156.25 | 161.25 | 118.50 | 125.00 | 1491.71 |
| 0-th | 185.00 | 115.00 | 123.33 | 153.33 | 78.57 | 90.29 | 139.75 | 107.00 | 105.75 | 132.50 | 1230.53 |
| 10-th | 185.00 | 115.00 | 118.33 | **168.33** | 81.63 | 89.71 | 136.00 | 102.75 | 108.00 | 110.00 | 1214.76 |
| 16-th | **195.00** | 145.00 | 138.33 | 165.00 | 143.54 | **136.76** | 157.75 | 166.00 | 118.50 | **140.00** | 1505.89 |
| 18-th | 190.00 | 145.00 | 133.33 | 160.00 | **144.56** | 136.47 | **159.75** | **167.75** | 121.50 | 130.00 | 1488.36 |
| 20-th | **195.00** | 150.00 | 133.33 | 165.00 | 142.52 | 134.12 | 157.00 | 165.25 | 115.75 | 132.50 | 1490.47 |
| 22-th | 190.00 | 150.00 | 143.33 | 165.00 | **144.56** | 134.41 | 156.25 | 167.5 | 118.25 | 132.50 | 1501.80 |
| 24-th | **195.00** | 153.33 | **148.33** | 165.00 | 142.52 | 134.71 | 157.00 | 163.75 | 116.25 | 132.50 | **1508.39** |

Table 1: Using LLaVA1.5 as the foundation model, the performance of **Activation Momentum** with different refinement starting layer on the MME dataset. Each column refers to a hallucination category and the best results in each column are marked with **bold**.

Here, $\eta$ represents the learning rate, and $\nabla L(\theta_{t-1})$ denotes the gradient w.r.t. the training objective. The term $v_t$ accumulates past gradients, serving as the *real updates* at the current step, while $\beta$ is the trade-off coefficient balancing historical and current information. This momentum-based approach is particularly effective in training deep neural networks, as it smooths the optimization trajectory, improving both stability and convergence speed (Sutskever et al., 2013).

**Activation Momentum in Vision-Language Models** Inspired by traditional momentum, we introduce the *Activation Momentum (AM)* in LVLMs, expecting to transfer its beneficial properties – guiding the change of hidden states toward a consistent and smooth update direction while preventing abrupt or localized deviations from historical updates. Specifically, given the hidden state $h_t^j$ at layer $j$ at time step $t$, the $j$-th transformer layer $f_j(\cdot)$ can produce

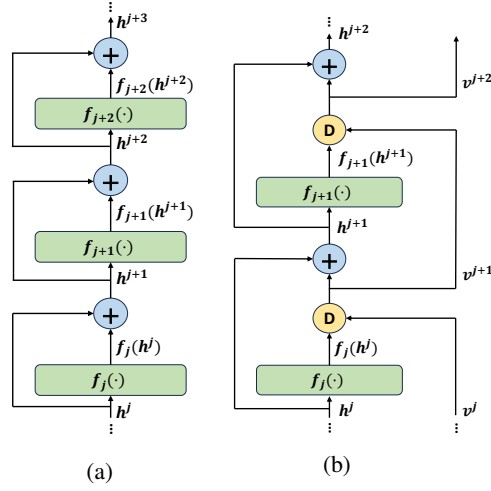

Figure 2: (a) The vanilla decoding scheme in LVLMs. (b) DAMO's decoding scheme.

the raw activation $\nabla h_t^{j+1} = f_j(h_t^j)$. In vanilla decoding process, the next-layer hidden state is obtained directly by $h_t^{j+1} = h_t^j + \nabla h_t^{j+1}$. To accumulate the previous activation status used to correct later layers, our proposed AM first integrates $\nabla h_t^{j+1}$ into the momentum term $v_t^j$. The momentum variable is formally defined as:

$$v_t^{j+1} = \beta v_t^j + (1 - \beta)\nabla h_t^{j+1}, \qquad (4)$$

where $\beta$, similar to its role in gradient descent, is the trade-off coefficient balancing historical and current information. And the term $v_t^{j+1}$ represents the momentum after incorporating the raw activation produced at the current step, which actually serves as the refined activation. Therefore, the revised updating process of hidden states is shown as below:

$$h_t^{j+1} = h_t^j + v_t^{j+1}, \qquad (5)$$

where $h_t^{j+1}$ can partially reflect the update trends of the previous layers and will then serve as the input to the subsequent layer. The aforementioned decoding process, involving the accumulation of activations momentum, is illustrated in Fig. 2b.

### 3.4 MECHANISM OF ADAPTIVE ACTIVATION REFINEMENT

**AM** represents a naive implementation of momentum-based decoding in LVLMs. However, unlike its use in optimization, we argue that while it is essential to maintain the momentum value (Equ. 4) all the time in order to preserve more historical information, Equ. 5 does not need to be applied across all layers. This is because hallucinations primarily emerge in the later layers so continuously refining the raw activations may suppress valuable updates encoding rich visual semantics generated by certain early layers.

To validate our proposition, we conduct a preliminary experiment on the MME dataset (Fu et al., 2023), a comprehensive benchmark to measure the ability of LVLMs in various aspects. The results shown in Table. 1 demonstrate that: ❶ Starting activation refinement too early (at layer 0 or 10)

significantly impairs the visual reasoning ability of LVLMs. ❷ Varying the starting layer for refinement can enhance different model capabilities (e.g., layer 16 excels in OCR tasks, while layer 24 improves positional perception). These observations suggest that activation refinement should begin in the later layers, and a fixed starting layer is insufficient to achieve comprehensive improvements across various key abilities of LVLMs.

Thus, we propose the *mechanism of adaptive activation refinement*, which also leverages the idea of momentum and adaptively determines, during forward computation, whether the model has reached the **hallucination surge** and selects the appropriate refinement starting layer. This mechanism, together with AM, constitutes our complete approach, **D**ecoding by **A**ccumulating Activations **MO**mentum (DAMO). For simplicity of notations, we denote the early exit distribution $p_j(x_{t+1}|x_{:t})$ at layer $j$ (from Equ. 2) as $P^j$. We focus on the update direction of the early exit distribution, maintaining momentum $\nabla \hat{P}^j$ to accumulate the update trends of the previous layers:

$$\begin{cases} \nabla P^j = P^j - P^{j-1}, \\ \nabla \hat{P}^j = \alpha \nabla \hat{P}^{j-1} + (1-\alpha)\nabla P^j, \end{cases} \tag{6}$$

where $\nabla P^j$ denotes the relative change of the early exit distribution caused by layer $j$, reflecting the current prediction tendencies of LVLMs. $\nabla \hat{P}^j$ is the momentum variable, preserving historical prediction trends of previous layers. Intuitively, if $\nabla P^j$ diverges significantly from the momentum variable $\nabla \hat{P}^j$, it indicates a notable deviation from the original trend, potentially signaling the onset of hallucination patterns. Based on this hypothesis, we use the cosine similarity between $\nabla P^j$ and $\nabla \hat{P}^j$ as the criterion for triggering activation refinement. Building on this hypothesis, we use the cosine similarity between $\nabla P^j$ and $\nabla \hat{P}^j$ as the criterion for triggering activation refinement. Specifically, if the similarity falls below a predetermined threshold $\tau$, activation refinement is initiated and continues until the final layer.

## 3.5 Coefficient Adjustment for Noise-Resistant Momentum

After we have reached the **hallucination surge**, keeping the trade-off coefficient $\beta$ (from Equ. 4) fixed leads to several issues: The hallucinated activations produced by the later layers will be incorporated into the activation momentum $v_t^j$, acting as noise and gradually diluting the correct historical information it contains. To address this, we set the initial value of $\beta$ to $\beta_1$ and change its value according to the following rules:

$$\beta = \begin{cases} \beta_2 & \text{if Cosine}(\nabla P^j, \nabla \hat{P}^j) < \tau \\ \beta_1 & \text{if Cosine}(\nabla P^j, \nabla \hat{P}^j) \geq \tau \end{cases} \tag{7}$$

Here we must ensure that $\beta_1 < \beta_2$. Transitioning from $\beta_1$ to $\beta_2$ means that the onset of a hallucination surge, accompanied by significant changes in prediction tendencies. Consequently, it becomes necessary to preserve more historical information (set a higher value for $\beta$, i.e. $\beta_2$), enhancing $v_t^j$, the momentum term's resistance to these hallucinatory elements.

## 4 Experiments

In experiments, we first introduced the datasets, models, and baselines in the Setup section, followed by a detailed presentation of the Results. We evaluated our DAMO method on the comprehensive hallucination dataset MME and examined object hallucination using the SEEM-annotated MSCOCO and A-OKVQA datasets from POPE. To assess robustness, we tested on the generalized dataset LLaVA-Bench. Additionally, we evaluated the adaptive activation refinement for starting layer selection and adaptive coefficient adjustment. Finally, we applied our DAMO to LLMs to assess its transferability to mitigate hallucinations.

### 4.1 Setup

**Datasets** We evaluate our proposed method using four datasets designed to assess hallucination issues in Large Vision-Language Models (LVLMs). MME Fu et al. (2023) offers a comprehensive

benchmark featuring 14 tasks categorized into perception and cognition. POPE (Polling-based Object Probing Evaluation) Li et al. (2023c) is a scalable framework for detecting object hallucinations in LVLMs, utilizing SEEM-annotated datasets from MSCOCO Lin et al. (2014) and A-OKVQA Schwenk et al. (2022). LLaVA-Bench, a generalized dataset, comprising 24 diverse images and 60 questions across categories like simple QA, detailed descriptions, and complex reasoning, is used to further evaluate the model's generalization and robustness.

**Models** Numerous high-performing LVLMs have emerged recently. For our evaluation, we selected three models: LLaVA1.5 Liu et al. (2023b), INF-MLLM1 Zhou et al. (2023a), and mPLUG-Owl2 Ye et al. (2023b), each demonstrating strong performance on established benchmarks. Notably, all these LVLMs are equipped with a 7B Large Language Models (LLMs).

**Baselines** We compare our method with four baselines. Regular responses are generated using the original LVLMs. Visual Contrastive Decoding (VCD) Leng et al. (2024) mitigates object hallucinations by contrasting output distributions from original and distorted visual inputs. DOLA Chuang et al. (2023) is transferred to LVLMs, with a fixed mature layer index at 32 and multiple candidate premature layer indices ("0, 2, 4, 6, 8, 10, 12, 14") to fine-grain the model's internal decision-making process. For OPERA Huang et al. (2024), due to the computational resource limitations, we set $num\_beams$ to 4, while keeping all other settings consistent with its official configuration. To ensure fairness, we set the temperature to 0 for all comparisons.

**Hyperparameters Setting** We have provided all the hyperparameters used in our experiments, including $\tau$, $\beta_1$, $\beta_2$ and $\alpha$, which can be found in the Appendix. Additionally, we tested the model's sensitivity to these hyperparameters, with detailed experimental results also available in the Appendix. The experimental results confirm that our hyperparameter selection is optimal.

## 4.2 RESULTS

**Evaluation on Comprehensive Hallucination Dataset**

▷ *Implementation* We adopted the MME benchmark to evaluate DAMO across 3 models and reported results in the perception category, which contains 10 tasks, following the settings used in other hallucination studies.

▷ *Q: Whether our DAMO outperforms other decoding methods on the comprehensive dataset?* Yes, as shown in Table 2, we present the experimental results of various decoding methods on the MME dataset. Notably, DAMO almost outperformed other decoding strategies across all models, achieving a total scores of 1515.89, 1520.74 and 1437.46 on LLaVA1.5, INF-MLLM1 and mPLUG-Owl2, respectively. Specifically, on the mPLUG-Owl2 model, DAMO surpassed the VCD method by 38.91 points. Moreover, in the "Celebrity" task, it achieved the highest score of 164.41 among all methods. Additionally, DOLA demonstrated no performance enhancement on the INF-MLLM1 model, suggesting that directly transferring DOLA to LVMs may not guarantee performance improvements.

▷ *Q: Does our DAMO also have relatively low memory consumption?* Yes, our DAMO also exhibits relatively low memory consumption. As shown in the last column of Table 2, the memory consumption of DAMO is nearly identical to that of Regular. In contrast, VCD requires two forward passes to contrast output distributions from original and distorted visual inputs, necessitating the storage of logits from both passes. DOLA also needs to retain logits from intermediate premature layers, while OPERA utilizes beam search, resulting in significantly higher memory consumption—up to 50GB on the INF-MLLM1 model. In summary, DAMO does not require storing any intermediate logits, as we continuously use momentum for modifications, which contributes to its efficiency in memory usage.

**Evaluation on Object Hallucination Dataset**

▷ *Implementation* To evaluate the effectiveness of our method in addressing object hallucinations, we compared with various decoding strategies on the SEEM-annotated MSCOCO and A-OKVQA datasets provided by POPE. We reported four key metrics: Accuracy, Precision, Recall and F1 Score, focusing primarily on Accuracy and F1 Score for brevity, with complete results available in the Appendix.

| Model | Decoding | Existence | Count | Position | Color | Posters | Celebrity | Scene | Landmark | Artwork | OCR | Total | Memory |
|---|---|---|---|---|---|---|---|---|---|---|---|---|---|
| LLaVA1.5 | Regular | 190.00 | **160.00** | 138.33 | 165.00 | 141.50 | 135.88 | 156.25 | 161.25 | 118.50 | 125.00 | 1491.71 | 14.6GB |
| | VCD | 188.33 | 140.00 | 133.33 | 155.00 | 137.76 | **139.12** | 153.25 | 166.00 | 120.75 | 125.00 | 1458.54 | 15.6GB |
| | DOLA | 190.00 | 158.33 | 143.33 | 165.00 | 139.46 | 133.24 | **157.75** | 160.50 | 119.50 | 125.00 | 1492.11 | 15.1GB |
| | OPERA | 195.00 | 158.33 | **148.33** | **175.00** | 142.52 | 131.18 | 157.00 | 161.50 | 117.00 | 132.50 | **1518.36** | 22.5GB |
| | DAMO | **195.00** | 150.00 | 143.33 | 165.00 | **144.56** | 135.00 | 157.00 | 166.00 | 120.00 | **140.00** | 1515.89 | **14.6GB** |
| INF-MLLM1 | Regular | 195.00 | 150.00 | 151.67 | 160.00 | 150.00 | 140.29 | 157.75 | 155.50 | 122.50 | 110.00 | 1492.71 | 17.6GB |
| | VCD | 190.00 | 138.33 | **160.00** | 160.00 | 142.52 | 135.59 | 157.00 | 153.50 | 113.25 | 105.00 | 1455.19 | 18.6GB |
| | DOLA | 195.00 | 150.00 | 151.67 | 160.00 | **150.00** | 140.29 | **157.75** | 155.50 | 122.50 | 110.00 | 1492.71 | 18.2GB |
| | OPERA | 195.00 | **155.00** | 151.67 | 160.00 | 149.32 | 139.41 | 156.25 | 154.00 | 122.50 | 110.00 | 1493.15 | 48.8GB |
| | DAMO | **195.00** | 150.00 | 158.33 | **165.00** | 148.64 | **146.76** | 156.25 | 154.75 | **128.50** | **117.50** | **1520.74** | **17.6GB** |
| mPLUG-Owl2 | Regular | 180.00 | 145.00 | 73.33 | 136.67 | 136.73 | 141.18 | 157.25 | 137.75 | 127.25 | 102.50 | 1337.66 | 16.1GB |
| | VCD | 185.00 | 155.00 | 63.33 | 148.33 | 142.86 | 158.53 | 154.00 | 141.00 | 133.00 | **117.50** | 1398.55 | 16.4GB |
| | DOLA | 190.00 | 160.00 | 70.00 | 150.00 | 145.92 | 159.12 | 160.50 | 151.25 | 130.25 | 110.00 | 1427.04 | 16.2GB |
| | OPERA | 190.00 | 160.00 | 70.00 | **150.00** | 145.92 | 160.88 | **160.50** | 150.50 | 131.00 | 110.00 | 1428.80 | 24.0GB |
| | DAMO | **190.00** | **160.00** | **75.00** | 145.00 | **148.30** | **164.41** | 157.50 | 155.75 | 131.50 | 110.00 | **1437.46** | **16.1GB** |

Table 2: Experimental results of various decoding strategies on MME dataset across three models: LLaVA1.5, INF-MLLM1 and mPLUG-Owl2.

▷ *Q: Does our proposed DAMO address object hallucinations effectively?* Yes, DAMO achieves significant performance improvements in both accuracy and F1 score, as evidenced by results from MSCOCO and A-OKVQA datasets.

❶ *MSCOCO dataset* Results from the MSCOCO dataset, as shown in Table 3, demonstrated the effectiveness of DAMO. In the popular setting with mPLUG-Owl2, DAMO improved accuracy from 80.73% to 82.77%, while F1 score increased from 82.52% to 83.80%. In the challenging adversarial setting, DAMO further enhanced accuracy and F1 score on LLaVA1.5 compared to Regular decoding, with improvements of 0.73% and 0.36%, respectively. In contrast, methods like VCD sometimes exhibited performance declines; for instance, in the popular setting, VCD's accuracy and F1 score dropped by 1.80% and 1.62% on LLaVA1.5 compared to Regular. The consistent superiority of DAMO reinforces the advantages of our approach.

❷ *A-OKVQA dataset* Turning to the A-OKVQA dataset shown in Table 3, DAMO again outperformed other decoding strategies. In the popular setting with mPLUG-Owl2, DAMO achieved significant improvements over the Regular decoding, with accuracy and F1 score increases of 3.30% and 2.07%, respectively, while other methods showed only minimal enhancements, with DOLA achieving improvements of merely 0.16% and 0.09%. The improvements on A-OKVQA were even more pronounced than those observed on MSCOCO, further underscoring the capability of our model.

**Evaluation on Generalized Dataset**

▷ *Implementation* To extend our evaluation beyond binary tasks, we conducted experiments on the generalized LLaVA-Bench dataset on LLaVA1.5, employing GPT-API for performance assessment.

▷ *Q: Can our proposed DAMO perform well on generalized dataset?* Yes, our DAMO demonstrated strong performance on the generalized LLaVA-Bench dataset. As shown in Table 4, our DAMO outperformed most decoding strategies across all categories. In the detailed description category, DAMO achieved a score of 75.87, surpassing DOLA by 1.20 points. It also excelled in complex reasoning, with a score of 88.39, which is significantly higher than OPERA's 69.64. With an overall score of 78.13, our DAMO demonstrated enhanced robustness and generalizability across diverse tasks, suggesting its potential for solving other challenges as well.

**Evaluation about Adaptive Activation Refinement**

▷ *Implementation* To fairly compare the effectiveness of adaptive starting layer selection strategies for activations refinement, we conducted several ablation experiments with fixed starting layers at the 0-th, 10-th, 16-th, 20-th, 24-th, and 28-th layers. All other parameters were kept consistent across experiments to ensure a fair comparison.

▷ *Q: Is our Adaptive Activation Refinement superior to the fixed layer starting for momentum?* Yes, the results presented in Figure 3a, obtained on the MME dataset with the LLaVA1.5 model, indicate that while fixed starting layers provide modest improvements over regular decoding, the adaptive layer selection consistently outperforms all fixed-layer approaches, demonstrating clear advantages by enabling a more responsive and effective decoding process.

**Evaluation about Adaptive Coefficient Adjustment**

▷ *Implementation* After determining the hallucination surge via adaptive activation refinement, we compared the performance of three setups: using only $\beta_1$, using only $\beta_2$ (without switching), and employing the adaptive switching strategy between $\beta_1$ and $\beta_2$. The experiments were conducted on

| | Dataset | | MSCOCO | | A-OKVQA | |
|---|---|---|---|---|---|---|
| Setting | Model | Decoding | Accuracy | F1 Score | Accuracy | F1 Score |
| Random | LLaVA1.5 | Regular | 89.63 | 89.74 | 87.30 | 88.49 |
| | | VCD | 87.53 | 87.81 | 85.00 | 86.49 |
| | | DOLA | 89.67 | 89.74 | 87.40 | 88.57 |
| | | OPERA | 89.87 | **89.95** | 87.27 | 88.50 |
| | | **DAMO** | **89.97** | 89.92 | **87.90** | **88.93** |
| | INF-MLLM1 | Regular | 91.17 | 90.92 | 90.60 | 90.97 |
| | | VCD | 90.00 | 89.65 | 89.87 | 90.21 |
| | | DOLA | 91.17 | 90.92 | 90.60 | 90.97 |
| | | OPERA | 91.27 | **91.02** | 90.57 | 90.94 |
| | | **DAMO** | **91.33** | 91.00 | **91.33** | **91.59** |
| | mPLUG-Owl2 | Regular | 86.27 | 86.88 | 81.57 | 83.89 |
| | | VCD | 84.40 | 84.79 | 82.53 | 84.16 |
| | | DOLA | 86.33 | 86.92 | 81.83 | 84.07 |
| | | OPERA | 86.23 | 86.84 | 81.53 | 83.86 |
| | | **DAMO** | **87.20** | **87.44** | **84.60** | **86.03** |
| Popular | LLaVA1.5 | Regular | 86.23 | 86.82 | 80.30 | 83.21 |
| | | VCD | 84.43 | 85.20 | 77.50 | 81.07 |
| | | DOLA | 86.20 | 86.75 | 80.47 | 83.32 |
| | | OPERA | 86.30 | 86.88 | 80.47 | 83.38 |
| | | **DAMO** | **86.70** | **87.07** | **81.27** | **83.84** |
| | INF-MLLM1 | Regular | 88.83 | 88.78 | 85.70 | 86.88 |
| | | VCD | 87.60 | 87.43 | 85.00 | 86.17 |
| | | DOLA | 88.83 | 88.78 | 85.70 | 86.88 |
| | | OPERA | 88.80 | 88.74 | 85.70 | 86.89 |
| | | **DAMO** | **89.30** | **89.11** | **86.67** | **87.62** |
| | mPLUG-Owl2 | Regular | 80.73 | 82.52 | 75.97 | 79.98 |
| | | VCD | 81.00 | 81.12 | 75.70 | 79.21 |
| | | DOLA | 80.87 | 82.60 | 76.13 | 80.07 |
| | | OPERA | 80.70 | 82.48 | 75.93 | 79.94 |
| | | **DAMO** | **82.77** | **83.80** | **79.27** | **82.05** |
| Adversarial | LLaVA1.5 | Regular | 79.70 | 81.71 | 69.33 | 76.10 |
| | | VCD | 78.13 | 80.38 | 67.90 | 75.01 |
| | | DOLA | 79.73 | 81.68 | 69.53 | 76.21 |
| | | OPERA | 79.77 | 81.77 | 69.20 | 76.09 |
| | | **DAMO** | **80.43** | **82.07** | **70.77** | **76.88** |
| | INF-MLLM1 | Regular | 84.87 | 85.38 | 76.13 | 79.88 |
| | | VCD | 84.17 | 84.47 | 76.37 | 79.82 |
| | | DOLA | 84.87 | 85.38 | 76.13 | 79.88 |
| | | OPERA | 84.87 | 85.36 | 76.10 | 79.85 |
| | | **DAMO** | **85.73** | **86.00** | **77.43** | **80.71** |
| | mPLUG-Owl2 | Regular | 76.17 | 77.69 | 67.37 | 74.63 |
| | | VCD | 77.10 | 77.00 | 68.80 | 74.85 |
| | | DOLA | 76.73 | 77.87 | 67.50 | 74.68 |
| | | OPERA | 76.87 | 78.01 | 67.30 | 74.58 |
| | | **DAMO** | **78.63** | **78.87** | **70.03** | **75.98** |

Table 3: Experimental results of various decoding strategies on the SEEM-annotated MSCOCO and A-OKVQA datasets from POPE using three models: LLaVA1.5, INF-MLLM1, and mPLUG-Owl2. The best values for each metric across all models and decoding strategies are highlighted in **bold**. the MME dataset using three models: LLaVA1.5, INF-MLLM1, and mPLUG-Owl2.with all other parameters kept constant across experiments.

▷ *Q: Does the adaptive momentum coefficient adjustment strategy outperform fixed coefficient setups?* Yes, the results presented in Figure 3b demonstrate that the adaptive $\beta$ switching strategy consistently outperformed the fixed-coefficient setups across all models. For instance, in the mPLUG-Owl2 model, using $\beta_1$ resulted in a total score increase of 48.66, while $\beta_2$ provided a larger increase of 90.49. In contrast, our adaptive approach achieved an impressive total score improvement of 99.8.

**Transferability Evaluation**

| Decoding | Conversation | Detail | Complex | Overall |
|----------|-------------|--------|---------|---------|
| Regular | 58.82 | 70.33 | 87.79 | 75.22 |
| VCD | 60.88 | 59.67 | 83.46 | 71.12 |
| DOLA | 61.47 | 74.67 | 88.04 | 77.17 |
| OPERA | **66.76** | 51.00 | 69.64 | 64.17 |
| DAMO | 63.24 | **75.87** | **88.39** | **78.13** |

Table 4: Comparisons about various decoding methods on LLaVA1.5 using LLaVA-Bench dataset.

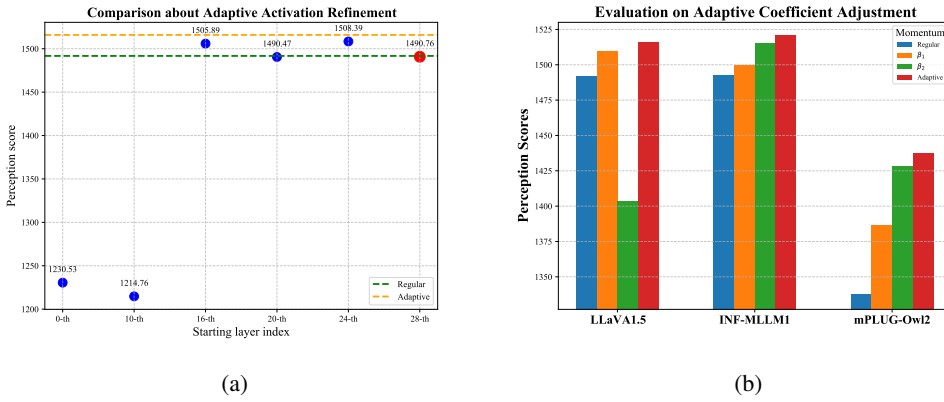

(a)                                                                              (b)

Figure 3: (a) Comparison of different starting layers for DAMO (0-th, 10-th, 16-th, 20-th, 24-th, and 28-th) against regular decoding and our adaptive activation refinement on the MME dataset using the LLaVA1.5 model. (b) Comparison results on adaptive momentum coefficient adjustment across LLaVA1.5, INF-MLLM1 and mPLUG-Owl2 on MME dataset. $\beta_1$ indicates the use of only $\beta_1$, $\beta_2$ denotes the use on only $\beta_2$, and *Adaptive* refers to adaptive adjustment between $\beta_1$ and $\beta_2$.

▷ *Implementation* To evaluate DAMO's transferability, we implemented it on the LLaMA2-7B model specifically targeting the effectiveness in addressing hallucinations in LLMs. The experiments focused on four datasets: TrthfulQA, FACTOR (News, Wiki), StrQA and GSM8K. We utilized Regular and DOLA decoding as baselines.

▷ *Q: Does our DAMO perform well when transferred to LLMs?* Yes, as shown in Table 5, DAMO improved performance on LLMs. In the TruthfulQA dataset, it outperformed Regular decoding in MC1 with a score of 34.15, surpassing both Regular (33.66) and DOLA (33.29). DAMO also excelled in the FACTOR task, scoring 57.48 on Wiki, higher than DOLA's 56.51. These results demonstrate DAMO's effectiveness when applied to LLMs, consistently improving performance across various language tasks.

| Decoding | TruthfulQA (MC) | | | FACTOR | | CoT | |
|----------|------|------|------|------|------|-------|-------|
| | MC1 | MC2 | MC3 | News | Wiki | StrQA | GSM8K |
| Regular | 33.66 | 51.29 | 24.91 | **65.44** | 56.91 | 63.67 | 21.25 |
| DOLA | 33.29 | **60.84** | **29.79** | 61.58 | 56.51 | **64.59** | **21.83** |
| DAMO | **34.15** | 51.24 | 24.95 | 64.67 | **57.48** | 63.67 | 21.30 |

Table 5: Performance comparison of different decoding methods, including our transferred momentum decoding, applied to the LLaMA2-7B model across various language tasks.

## 5 CONCLUSION

In this paper, we introduced DAMO, a momentum decoding method aimed at mitigating hallucinations in Large Vision-Language Models (LVLMs) by accumulating visual information from earlier layers, where we found that correct information often appears in the early stage. By refining activations throughout the inference procedure, DAMO effectively preserves essential visual semantics, leading to more accurate and reliable predictions. Our method achieved excellent results on the MME, POPE, and LLaVA-Bench datasets, showcasing its effectiveness in visual-language benchmarks. Furthermore, when transferred to Large Language Models (LLaMA2), DAMO maintained strong performance across several datasets, including TruthfulQA, FACTOR (News, Wiki), StrQA, and GSM8K. These findings validate the robustness of our approach, underscoring its potential to enhance the reliability of models across a wide range of tasks.

ACKNOWLEDGEMENTS

This work is, in part, supported by NSF (#CNS2431516). We thank the anonymous reviewers for their constructive feedback on this work. The views, opinions, and/or findings contained in this paper are those of the authors and should not be interpreted as representing any funding agencies.

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

## A  HYPERPARAMETER SETTING

We provide our detailed hyperparameter setting in Table 6. $\alpha$ is set to 0.7 for all experiments. And we present different hyperparameters including $\tau$, $\beta_1$ and $\beta_2$ for different tasks across different models. M represents MME dataset and P denotes POPE dataset.

|  | LLaVA M | LLaVA P | INF M | INF P | mPLUG M | mPLUG P |
|---|---|---|---|---|---|---|
| $\beta_1$ | 0.05 | 0.20 | 0.20 | 0.05 | 0.60 | 0.40 |
| $\beta_2$ | 0.20 | 0.40 | 0.40 | 0.10 | 0.80 | 0.60 |
| $\tau$ | -0.30 | -0.30 | -0.60 | -0.60 | -0.30 | -0.30 |

Table 6: Hyperparameters for different tasks across different models

## B  ABLATION STUDIES

### B.1  EVALUATING THE SENSITIVITY OF DIFFERENT $\tau$

We evaluate the sensitivity of different $\tau$ on POPE (MSCOCO setting) dataset using LLaVA1.5 model and report F1 score. As shown in Table 7, by setting $\tau$ to 0.3, 0, -0.3, and -0.6, we observe that the model's performance varies accordingly. This demonstrates the model's sensitivity to $\tau$. Notably, when $\tau$ is set to our default value of 0.3, the model achieves its best performance.

| $\tau$ | 0.3 | 0 | -0.3 | -0.6 |
|---|---|---|---|---|
| Random | 88.83 | 89.56 | **89.92** | 89.62 |
| Popular | 86.60 | 86.43 | **87.07** | 86.66 |
| Adversarial | 81.95 | 81.56 | **82.07** | 81.78 |

Table 7: The sensitivity of $\tau$ on POPE (MSCOCO) dataset using LLaVA1.5 model.

### B.2  EVALUATING THE SENSITIVITY OF DIFFERENT $\beta_1$

We evaluate different $\beta_1$ values to observe performance variations on the MME dataset using LLaVA1.5 model. For a fair comparison, $\beta_2$ is set to its default value of 0.2. In this experiment, we explore setting $\beta_1$ to 0.01, 0.05, 0.1, 0.15, and 0.2, and analyze the corresponding results.

As shown in Table 8, LLaVA1.5 is quite sensitive to the value of $\beta_1$ . When $\beta_1$ is set to smaller values, the model shows improved performance compared to the baseline. However, as $\beta_1$ increases to 0.20, a noticeable performance drop is observed. This highlights that 0.05 is the optimal parameter setting.

| $\beta_1$ | Existence | Count | Position | Color | Posters | Celebrity | Scene | Landmark | Artwork | OCR | Total |
|---|---|---|---|---|---|---|---|---|---|---|---|
| 0.01 | 190.00 | 145.00 | 135.00 | 160.00 | 149.66 | 137.35 | 160.00 | 168.50 | 119.75 | 137.50 | 1502.76 |
| 0.05 | 195.00 | 150.00 | 143.33 | 165.00 | 144.56 | 135.00 | 157.00 | 166.00 | 120.00 | 140.00 | **1515.89** |
| 0.10 | 195.00 | 151.67 | 153.33 | 170.00 | 136.39 | 130.00 | 155.50 | 160.50 | 117.75 | 125.00 | 1495.14 |
| 0.15 | 190.00 | 153.33 | 143.33 | 180.00 | 125.51 | 122.94 | 151.00 | 150.75 | 113.00 | 132.50 | 1462.37 |
| 0.20 | 190.00 | 120.00 | 145.00 | 175.00 | 122.45 | 120.00 | 145.25 | 136.00 | 110.25 | 140.00 | 1403.95 |

Table 8: Comparison of different $\beta_1$ on the MME dataset using LLaVA1.5 model.

### B.3  EVALUATING THE SENSITIVITY OF DIFFERENT $\beta_2$

Similar to the evaluation of $\beta_1$ , we fix $\beta_1$ at 0.05 and evaluate the performance across different $\beta_2$ values: 0.1, 0.2, 0.4, 0.6, and 0.8.

| $\beta_2$ | Existence | Count | Position | Color | Posters | Celebrity | Scene | Landmark | Artwork | OCR | Total |
|---|---|---|---|---|---|---|---|---|---|---|---|
| 0.10 | 195.00 | 156.67 | 143.33 | 170.00 | 138.44 | 133.53 | 156.25 | 160.50 | 118.75 | 125.00 | 1497.46 |
| 0.20 | 195.00 | 150.00 | 143.33 | 165.00 | 144.56 | 135.00 | 157.00 | 166.00 | 120.00 | 140.00 | **1515.89** |
| 0.40 | 185.00 | 130.00 | 133.33 | 143.33 | 137.41 | 123.82 | 159.25 | 160.00 | 110.00 | 102.50 | 1384.66 |
| 0.60 | 188.33 | 103.33 | 93.33 | 120.00 | 123.81 | 107.94 | 132.75 | 124.00 | 88.25 | 105.00 | 1186.75 |
| 0.80 | 183.33 | 113.33 | 91.67 | 135.00 | 121.09 | 102.35 | 127.75 | 121.50 | 85.25 | 122.50 | 1203.77 |

Table 9: Comparison of different $\beta_2$ on the MME dataset using LLaVA1.5 model.

As shown in Table 9, the model's performance is significantly affected by changes in $\beta_2$. When $\beta_2$ is set too high, the model's performance rapidly declines, as this interferes with the model's normal reasoning process. Our experiments also confirm that the default setting of $\beta_2$ at 0.2 is optimal.

### B.4 EVALUATING THE EFFECT OF OUR PROPOSED ADAPTIVE ACTIVATION REFINEMENT MECHANISM

In addition to testing the effectiveness of our adaptive activation refinement mechanism on LLaVA1.5, we further evaluate it on INF-MLLM1 by setting the starting layer to the 16th, 20th, 24th, and 28th layers.

| | Existence | Count | Position | Color | Posters | Celebrity | Scene | Landmark | Artwork | OCR | Total |
|---|---|---|---|---|---|---|---|---|---|---|---|
| 16-th | 190.00 | 150.00 | 158.33 | 165.00 | 146.94 | 147.65 | 157.00 | 150.75 | 126.75 | 117.50 | 1509.92 |
| 20-th | 190.00 | 150.00 | 158.33 | 165.00 | 145.92 | 147.65 | 156.25 | 150.00 | 128.50 | 125.00 | 1516.65 |
| 24-th | 185.00 | 145.00 | 163.33 | 165.00 | 142.86 | 149.12 | 154.75 | 147.75 | 128.25 | 132.50 | 1513.56 |
| 28-th | 190.00 | 140.00 | 158.33 | 165.00 | 143.88 | 148.82 | 154.75 | 147.00 | 127.25 | 132.50 | 1507.53 |
| Adaptive | 195.00 | 150.00 | 158.33 | 165.00 | 148.64 | 146.76 | 156.25 | 154.75 | 128.50 | 117.50 | **1520.74** |

Table 10: Comparison of different momentum decoding starting layers (16-th, 20-th, 24-th, and 28-th) against regular decoding and our adaptive layer selection on the MME dataset using the INF-MLLM1 model.

As shown in Table 10, our adaptive starting layer method outperforms all fixed starting layer settings, further demonstrating the effectiveness of our approach.

