# OpenReview forum: "DAMO: Decoding by Accumulating Activations Momentum for Mitigating Hallucinations in Vision-Language Models"
_ICLR.cc/2025/Conference — ICLR 2025 Poster_

### Official Review · Reviewer_J6Q4 · 2024-10-26

**Soundness:** 3
**Presentation:** 2
**Contribution:** 2
**Rating:** 6
**Confidence:** 4

**Summary:**

This paper addresses the issue of hallucinations in Large Vision-Language Models (VLMs) by analyzing their layer-wise prediction tendencies and decoding mechanisms.  The authors propose a decoding strategy inspired by the momentum concept in gradient descent optimizers, which ensures adaptive decoding consistency across layers.

**Strengths:**

1. The authors conduct a detailed layer-wise analysis of VLMs' prediction tendencies, identifying that hallucinations often emerge in the final layers due to significant shifts in decoding. This insight may provide a deeper understanding of the underlying mechanisms causing hallucinations.

2. The proposed Decoding by Accumulating Activations MOmentum (DAMO) method is somewhat effective. By accumulating momentum across layers, DAMO ensures consistent decoding and reduces the impact of late-stage hallucinations.

**Weaknesses:**

1. The paper draws inspiration from the momentum concept and contrast decoding, but it lacks a detailed analysis and comparison with recent related work, such as [1, 2]. Additionally, the paper does not adequately explain why the proposed method theoretically offers advantages over previous approaches.

2. Poor presentation: The paper lacks comprehensive framework diagrams that clearly illustrate the specific content and workflow of the proposed method, making it difficult to understand the method's details.

3. The paper does not provide a detailed analysis of how sensitive DAMO is to hyperparameters, such as the momentum coefficient. Understanding the optimal settings for different models and tasks could help in practical implementation.


[1] Alleviating Hallucinations in Large Vision-Language Models through Hallucination-Induced Optimization

[2] Ibd: Alleviating hallucinations in large vision-language models via image-biased decoding

**Questions:**

Please refer to Weakness

---

> ### Author Response · Authors · 2024-11-20
> **Thank you! (1/2)**
>
> We sincerely appreciate your review and the valuable suggestions provided.
> # W1 - More comparison with recent related work
> *  We appreciate your suggestions regarding the two recent related works. We have revised the related work including detailed comparison with these two excellent works. You could check the updated version. And we will perform a comparison with them in this section.
> *  Similar to DoLA's method, IBD ((Image-Biased Decoding) generates the next-token probability distribution by comparing predictions from a standard LVLM with those from an image-biased LVLM, amplifying image-related information and reducing hallucinations from over-reliance on text. In another perspective, our preliminary experiments show that VLMs can extract image features, but hallucinations arise during inference. So DAMO addresses this by modifying the hidden states across layers to correct the inference. We believe that the problems and objectives addressed by our method are fundamentally different from those of IBD.
> * For HIO (Hallucination-Induced Optimization), it is also a contrasting method essentially. HIO addresses hallucinations in Large Visual Language Models (LVLMs) by using a fine-tuned Contrary Bradley-Terry Model to enhance the contrast between hallucinatory and correct tokens. However, HIO requires fine-tuning an Evil LVLM for contrasting decoding, adding extra training costs. In contrast, DAMO solves hallucinations in VLMs by modifying hidden states during inference, without the need for contrasting decoding and auxiliary training. This avoids the issues about contrasting decoding as illuatrated in HIO, such as the uncontrollable global visual uncertainty, which can cause imprecise hallucination detection and lead to undesired hallucinations.
>  * In summary, unlike previous methods that rely solely on the logits from the final layer for contrastive decoding, our approach focuses on the variations in hidden states during the inference process. By analyzing changes in the words corresponding to the hidden states, we first detect hallucinations through cosine similarity. Then we apply a momentum-like method to correct the hidden states, aiming to keeping visual semantics consistent. I think this could address your concerns about why the proposed method theoretically offers advantages over previous approaches.
>
> # W2 - More comprehensive framework diagrams are needed: We have presented a new diagram in Appendix.
> * Sorry for this confusion. We have drawn a diagram to further illustrate our approach. You could check in Appendix(I). We compare the inference process between traditional VLMs and our DAMO's modifying hidden states. Hope this could address your concerns. Thanks again for your valuable advice.

---

> > ### Author Response · Authors · 2024-11-22
> > **Thank you! (2/2)**
> >
> > # W3 -  More detailed analysis of how sensitive DAMO is to hyperparameters:
> > * We have released all hyperparameters in Appendix (D. Hyperparameter Setting). $\alpha$ is set to 0.7 for all experiments. And we present different hyperparameters including $\tau$, $\beta_1$ and $\beta_2$ for different tasks across different models. M represents MME dataset and P denotes POPE dataset.
> > |   |   LLaVA1.5 M  |  LLaVA1.5 P   |  INF-MLLM1 M   | INF-MLLM1 P    | mPLUG-Owl2 M | mPLUG-Owl2 P |
> > | --------- | --- | --- | --- | --- | -------- | -------- |
> > | $\beta_1$ |  0.05   |   0.20  |  0.20   |  0.05   |       0.60   |      0.40    |
> > |    $\beta_2$  | 0.20    |   0.40  |  0.40   |   0.10  |     0.80     |     0.60     |
> > |   $\tau$   |  -0.30   |   -0.30  | -0.60    |   -0.60  | -0.30     | -0.30     |
> >
> > * We first evaluate the sensitivity of different $\tau$ on POPE (MSCOCO setting) dataset using LLaVA1.5 model and report F1 score. As shown in next Table, by setting $\tau$ to 0.3, 0, -0.3, and -0.6, we observe that the model's performance varies accordingly. This demonstrates the model's sensitivity to $\tau$. Notably, when $\tau$ is set to our default value of 0.3, the model achieves its best performance. You could check details in Appendix (E.1)
> > | $\tau$      | 0.3  | 0   |   -0.3  | -0.6 |
> > | ----------- | ---- | --- | --- | ---- |
> > | Random      |   88.83 |  89.56  | **89.92** |  89.62         |
> > | Popular     |  86.60|   86.43|  **87.07** |86.66 |
> > | Adversarial |81.95|  81.56  |**82.07**| 81.78 |
> >
> > * We then evaluate different $\beta_1$  values to observe performance variations on the MME dataset using LLaVA1.5 model. For a fair comparison, $\beta_2$ is set to its default value of 0.2. In this experiment, we explore setting $\beta_1$ to 0.01, 0.05, 0.1, 0.15, and 0.2, and analyze the corresponding results. As shown in next Table, LLaVA1.5 is quite sensitive to the value of $\beta_1$ . When $\beta_1$ is set to smaller values, the model shows improved performance compared to the baseline. However, as $\beta_1$ increases to 0.20, a noticeable performance drop is observed. This highlights that 0.05 is the optimal parameter setting. You could check details in Appendix (E.2)
> >  |   $\beta_1$  |  0.01   |  0.05   | 0.1 | 0.15 | 0.2 |
> > | --- | --- | --- | -------- | -------- | -------- |
> > |   MME score  |   1502.76  |   **1515.89**  | 1495.14 | 1462.37   | 1403.95    |
> >
> > *  We then evaluate different $\beta_2$  values to observe performance variations on the MME dataset using LLaVA1.5 model. Similar to the evaluation of $\beta_1$ , we fix $\beta_1$ at 0.05 and evaluate the performance across different $\beta_2$  values: 0.1, 0.2, 0.4, 0.6, and 0.8. As shown in next Table, the model's performance is significantly affected by changes in $\beta_2$. When $\beta_2$ is set too high, the model's performance rapidly declines, as this interferes with the model's normal reasoning process. Our experiments also confirm that the default setting of $\beta_2$ at 0.2 is optimal.
> > | $\beta_2$ | 0.10 |    0.20 | 0.40    | 0.60    | 0.80 |
> > | -------- | -------- | --- | --- | --- | -------- |
> > | MME score     | 1497.46| **1515.89**    |   1384.66  |   1186.75  | 1203.77 |
> > * We further evaluate our proposed adaptive activation refinement mechanism.  In addition to testing the effectiveness of our adaptive activation refinement mechanism on LLaVA1.5, we further evaluate it on INF-MLLM1 by setting the starting layer to the 16th, 20th, 24th, and 28th layers. As shown in the next Table, our adaptive starting layer method outperforms all fixed starting layer settings, further demonstrating the effectiveness of our approach.
> > | Starting Layer | 16-th |   20-th  |  24-th   |  28-th   | Adaptive |
> > | -------- | -------- | --- | --- | --- | -------- |
> > | MME score     | 1509.92     |  1516.65   |  1513.56   |  1507.53   | **1520.74** |
> >
> >
> >
> > Hope these explanations could address your concerns. Please let me know if you have any question.

---

> > > ### Comment · Reviewer_J6Q4 · 2024-11-25
> > >
> > > Thanks for your detailed responses, which address my concerns. I increase the score accordingly

---

> > > > ### Author Response · Authors · 2024-11-25
> > > >
> > > > Thanks for your valuable feedback!!!

---

### Official Review · Reviewer_XzDy · 2024-10-29

**Soundness:** 3
**Presentation:** 2
**Contribution:** 3
**Rating:** 6
**Confidence:** 4

**Summary:**

This paper investigates the layer-wise prediction tendencies of VLMs and proposes a novel decoding strategy called DAMO to mitigate hallucinations. DAMO enforces decoding consistency across layers in an adaptive manner during forward passes, amplifying visual semantics consistently extracted throughout the inference while reducing hallucination biases. Experiments on various VLM benchmarks demonstrate that DAMO significantly mitigates hallucinations, resulting in more visually grounded and accurate predictions.

**Strengths:**

1. Novel insights consider the layer-wise outputs, especially regarding to language bias in LVLMs.

2. Simple but effective solutions named DAMO based on the proposed findings.

3. Extensive experiments and pleasant results are achieved.

4. Source codes and partial experimental results are provided.

**Weaknesses:**

1. Although this method is proposed for LVLMs, this paper does not mention the DoLa [r1] in the main body. DoLa's claims are somewhat opposite to DAMO's. More analysis is needed to clarify the insights and differences.
[r1] Dola:  Decodingbycontrastinglayers improves factualityinlarge languagemodels. arXivpreprint
 arXiv:2309.03883,2023.
2. Applying DAMO to LVLMs and LLMs seems to be contradictory, especially considering the DoLa's findings.
3. I check the appendix and think some critical experimental details of DAMO are missing, like temperature, max token, decoding strategies.
4. GPT4-V experiments should be better conducted rather than GPT4, following VCD.
5. typo: INF-MLLM1 $\rightarrow$ INF-MLLM.

**Questions:**

1. How about some complex reasoning VQA performances in terms of Figure 1(b) (c). As indicated in DoLa [r1], deep layers facilitate the complex reasoning. Also, the reasoning steps is more complex as in [r1], please add more explanations.
[r1] Dola:  Decodingbycontrastinglayers improves factualityinlarge languagemodels. arXivpreprint
 arXiv:2309.03883,2023.
2. In 'Figure 1(c) Proportion of samples in a small dataset....',  Detailed descriptions in the Figure 1 caption are better illustrated.
3. Better re-organize Figure 1 to save more space.

I consider adjusting my score based on the clarification of DAMO and DoLa.

---

> ### Author Response · Authors · 2024-11-20
> **Thank you! (1/2)**
>
> We sincerely appreciate your review and the valuable suggestions provided.
>
> # W1 & W2 More comparisons and analysis about DoLA and DAMO: Thank you for your feedback and for highlighting DoLa's excellent work. Indeed, our approach was inspired by DoLa to some extent.
> * We have compared with DoLa in the updated version and will compare Dola more in camera-ready version. We will illustrate the differences here.
> * While DoLA is applied to LLMs and uses contrasting layers probabilities to better surface factual knowledge and reduce the generation of incorrect facts, our DAMO approach is designed for VLMs. DAMO focuses more on visual semantics rather than factual information obtained during model training, making it fundamentally different in its objectives and application. Since DoLA and DAMO aim to address different problems, we believe it is reasonable that the patterns they exhibit differ accordingly.
> * As depicted in Figure 1 of DoLA, the probabilities for "Seattle" remain relatively stable across different layers, while the probability for the correct answer, "Olympia," progressively increases from lower to higher layers. DoLa utilizes this observation by contrasting the differences between layers to enhance the model’s probability towards factually accurate outputs. On the other hand, DAMO identifies two distinct stages in the inference process concerning visual information, as shown in Figure 1: Visual Information Extraction and Loss of Visual Information. So we proposed DAMO to keep these visual semantics consistent. We believe these findings are not contradictory, as they aim to solve different issues in their respective contexts.
>
> * In summary, unlike DoLA-like methods that rely  on the logits from the final layer for contrastive decoding, our approach focuses on the variations in hidden states during the inference process. By analyzing changes in the words corresponding to the hidden states, we first detect hallucinations through cosine similarity. Then we apply a momentum-like method to correct the hidden states, aiming to keeping visual semantics consistent.
>
> * We have also experimentally confirmed that the comparison between DoLA and DAMO makes sense. First, in experiments on VLMs using the MME dataset, POPE dataset, and LLaVA-Bench dataset, we found that DAMO outperforms DoLA, as shown in Tables 2, 3, and 4. Additionally, we conducted experiments on LLMs using three different datasets: TruthfulQA (MC), FACTOR, and CoT. These experiments confirmed that DoLA almost performs better than DAMO on LLMs to some extent, as shown in Table 5. Therefore, we believe that DoLA and DAMO are not contradictory; they both perform well in their respective domains, with DoLA excelling in LLMs and DAMO performing better in VLMs.
>
> # W3 - Experimental details missing: We have added them in the experiment section in the updated version.
> * Thank you for your advice. We have released our complete set of hyperparameters. For a fair comparison, we set the `temperature` to `0` and the `max_token` to `1024` for all experiments. For decoding strategies, `regular decoding` refers to `greedy search`, while all other methods follow their original settings. The settings for other decoding strategies are described in lines 324-331. Furthermore, for all hyperparameters used in our experiment, you could check in Appendix (D).
> |   |   LLaVA M  |  LLaVA P   |  INF M   | INF P    | mPLUG M | mPLUG P |
> | --------- | --- | --- | --- | --- | -------- | -------- |
> | $\beta_1$ |  0.05   |   0.20  |  0.20   |  0.05   |       0.60   |      0.40    |
> |    $\beta_2$  | 0.20    |   0.40  |  0.40   |   0.10  |     0.80     |     0.60     |
> |   $\tau$   |  -0.30   |   -0.30  | -0.60    |   -0.60  | -0.30     | -0.30     |
>
> # W4 - GPT4-V experiments are better: We will reconduct in the camera-ready version.
> * Thank you for your suggestion. We followed the evaluation methodology outlined by the official LLaVA-Bench. We plan to update our experiments using GPT-4V in the camera-ready version.
>
> # W5 -  typo: INF-MLLM1 --> INF-MLLM: We will clarify it in detail.
> * Thanks for pointing that out. INF-MLLM includes two versions: INF-MLLM1 titled ` InfMLLM: A Unified Model for Visual-Language Tasks` and INF-MLLM2 titled `INF-MLLM2: High-Resolution Image and Document Understanding`. These versions are named as such in their official repository. In our work, we specifically adopted INF-MLLM1 as the backbone model.

---

> ### Author Response · Authors · 2024-11-20
> **Thank you! (2/2)**
>
> # Q1 -  Complex reasoning VQA performances: We present an example to elaborate this question.
> * Although our methods aim to keep visual semantics consistent across layers during inference, we are pleased to present an example to illustrate the complex reasoning. You could check in Appendix (G).
> * The results demonstrate that our method correct the hidden states during inference when trying to answer complex reasoning questions. Given an animated image and the question `“Are all the characters in this figure from 'The Story of the Little Mole?' ”`, the ground truth is that ` No, “Jerry the Mouse” is from the popular cartoon “Tom and Jerry” and not from “The Story of the Little Mole” `. For fair comparison, we set temperature to 0, and the results are as follows:
>
>     Regular Decoding: `No, not all the characters in the figure are from The Story of the Little Mole. The figure includes a variety of characters, such as a mouse, a hedgehog, a frog, a turtle, and a rabbit, as well as a character from The Story of the Little Mole. `
>
>     VCD : `No, not all the characters in the figure are from The Story of the Little Mole. The characters include a mouse, a frog, a hedgehog, a mole, and a squirrel, but there is also a character from Paddington Bear.`
>
>     OPERA: `No, not all the characters in the figure are from The Story of the Little Mole. The figure includes a variety of characters, such as a mouse, a hedgehog, a frog, a turtle, and a rabbit, as well as a character from The Story of the Little Mole.`
>
>     DoLA: `No, not all the characters in the figure are from The Story of the Little Mole. The figure includes a variety of characters, such as a mouse, a hedgehog, a frog, a turtle, and a rabbit, as well as a character from The Story of the Little Mole.`
>
>     DAMO: `No, not all the characters in the figure are from The Story of the Little Mole. Jerry the Mouse is from the popular cartoon ``Tom and Jerry," and he is not from ``The Story of the Little Mole." The other characters in the figure are from "The Story of the Little Mole."`
>
> The results show that Regular Decoding could not decode correct answer and OPERA and DoLA generate answers consistent with Regular Decoding. Only DAMO and VCD generate answers different from Regular Decoding. Furthermore, VCD generates incorrect answer that `There is no character from Paddington Bear`,  only DAMO address the hallucination in this example, decoding correct answer that `“Jerry the Mouse” is from the popular cartoon “Tom and Jerry”`.  We believe that this example could demonstrate DAMO's ability in complex reasoning.
>
> # Q2 - More descriptions in the Figure 1:we have revised this issue in the updated version.
> * Original Figure 1(c) aims to illustrate that ``Proportion of samples (75%) from COCO val2014, where questions about quantifiable attributes (e.g., color, number, direction) reveal that visual information is extracted in early layers but lost in later layers, leading to hallucinations.``
> * We abstained Figure 1(c) in the updated version. More details about Figure 1(c) has been illustrated in our motivation part.
>
> # Q3 Better re-organize Figure 1 to save more space: We have revised our Figure 1(c).
>
> * We have redrawn Figure 1. You could check in the updated version.
>
> Hope these explanations could address your concerns. Please let me know if you have any questions.

---

> > ### Comment · Reviewer_XzDy · 2024-11-22
> > **Thanks for detailed response!**
> >
> > Thanks for detailed response! I will read the whole comments and responses.

---

> > > ### Author Response · Authors · 2024-11-22
> > >
> > > Sure, please let me know if you have any question. I am really pleased to provide more information you need.

---

> > > > ### Comment · Reviewer_XzDy · 2024-11-23
> > > > **Official Comment by Reviewer XzDy**
> > > >
> > > > Thanks for the replay. I maintain the positive score and increase the confidence.

---

> > > > > ### Author Response · Authors · 2024-11-24
> > > > >
> > > > > Thanks for your reply. We are really pleased to response any question from you. Please let us know if you have any additional question.

---

### Official Review · Reviewer_5XME · 2024-11-03

**Soundness:** 4
**Presentation:** 2
**Contribution:** 3
**Rating:** 6
**Confidence:** 5

**Summary:**

The paper proposes DAMO (Decoding by Accumulating Activations Momentum), a novel decoding strategy designed to reduce hallucinations in Vision-Language Models (VLMs) by maintaining consistency across layers during inference. DAMO introduces a momentum-based mechanism in transformer forward computation to smooth activation updates layer-by-layer, mitigating hallucinations by preserving visual grounding in predictions. DAMO shows good results on various benchmarks (MME, POPE) and models (LLaVA1.5, INF-MLLM1, mPLUG-Owl2) with minimal memory overhead. Its adaptability is demonstrated through successful application to LLM benchmarks such as TruthfulQA and FACTOR, highlighting DAMO’s transferability across various tasks.

**Strengths:**

- The paper presents an interesting observation that VLMs are often able to generate accurate predictions in the initial layers, with hallucinations surfacing in later layers. This suggests that hallucinations could stem from disruptions in the decoding sequence rather than an absence of underlying knowledge, providing a novel diagnostic angle on VLM hallucinations.
- The paper’s approach to using "momentum" within the forward computation of transformers is intuitive and impactful. By building consistency across activations, DAMO effectively mitigates hallucination emergence in later layers, reinforcing layer-wise stability and preserving visual grounding.
- The authors extend their method beyond VLMs to yield strong performance in LLM benchmarks like TruthfulQA, FACTOR, StrQA and GSM8K, showcasing its versatility across various language tasks.

**Weaknesses:**

- While the Q&A format in the experiments section is engaging, it’s overused, which might dilute its effectiveness. Reserving this style for only the most critical insights could enhance focus and readability.
- Although DAMO shows good results on the MME benchmark, the gains seem modest for hallucination evaluation. Emphasizing the "celebrity" score on mPLUG-Owl2 might not strongly correlate with reduced hallucinations. The MME benchmark may be more reflective of general VLM performance rather than specific hallucination robustness. To strengthen authors’ claims on hallucination mitigation, it would be beneficial to test its performance on other benchmarks that more directly target hallucination evaluation.
- The paper introduces a threshold \tau as a cosine similarity criterion for switching between coefficients \beta_1​ and \beta_2​, which is critical to DAMO’s performance. However, the manuscript lacks explanation on how \tau was determined, its specific value, and its impact on model effectiveness. Additional details would improve understanding and reproducibility.

**Questions:**

- The results on the MME dataset show only modest improvements in hallucination reduction, with much of the gains appearing in tasks that may be more knowledge-based (e.g., Celebrity, Posters, Landmark, Artwork) rather than directly related to hallucination robustness. Would you consider testing DAMO on additional hallucination-focused benchmarks? Alternatively, could you clarify how the MME dataset captures DAMO's effectiveness in mitigating hallucinations?
- Given that the threshold τ\tauτ is central to controlling the switch between coefficients \beta_1​ and \beta_2​, could you provide more information on how its value was determined? Additionally, could you discuss any observed sensitivity of DAMO’s performance to variations in \tau?
- Could you provide further insights on the actual layers where the adaptive coefficient adjustment activates? For instance, does the transition of \beta from \beta_1​ to \beta_2​ tend to occur around specific layers, such as the after 24th, as observed in your analysis? Clarifying whether these transitions align with the predefined analysis layers would help in understanding DAMO’s consistency and effectiveness.

---

> ### Author Response · Authors · 2024-11-20
> **Thank you! (1/1)**
>
> We sincerely appreciate your review and the valuable suggestions provided.
>
> # W1 - Q&A format in the experiments section is overused: We will revise it in the camera-ready version.
>
> * Thanks for your advice. After combined more experiments we conducted, we will revise it in the camera-ready version to only stress the most critical insights.
>
> # W2 & Q1 - More experiments on hallucination-targetd benchmarks: We have evaluate DAMO on another hallucination benchmark named HallusionBench.
>
> * Due to the inconsistent experimental settings in previous works using CHAIR (e.g., some studies randomly selected images for comparison), we have used a new hallucination benchmark called HallusionBench for evaluation. HallusionBench offers a well-established evaluation framework that facilitates consistent and reliable experimentation. We present results here and we believe this results could demonstrate our DAMO's effectiveness on hallucination-targeted benchmarks. Details can be found in Appendix (H).
> | Model     | qAcc   | fAcc    | easy aAcc | hard aAcc | aAcc    |
> | --------- | ------ | ------- | --------- | --------- | ------- |
> | LLaVA1.5  | 8.13 | 14.16 | 36.92   | 27.91    | 35.43 |
> | VCD       | 9.01  | 14.16 | 36.70   | 29.53   | 35.96 |
> | DoLA      | 9.45 | 14.45 | 36.70   | 29.07   | 36.23 |
> | OPERA     | 8.57 | 14.45 | 37.14   | 29.07   | 35.96  |
> | DAMO      |   **9.67**     | **14.74** |**37.14**|**29.77**     |  **36.58**       |
> | INF-MLLM1 | 7.69 | **14.45** | 41.98    | 29.07   | 38.62 |
> | VCD       | 7.25 | 13.01 | 40.66  | 28.37   | 37.73 |
> | DoLA      | 7.03 |  13.01 | 40.22  | 29.30|38.00 |
> | OPERA     | 7.25|  13.87| 41.76| 27.44 | 37.91 |
> | DAMO      |     **8.13** |  14.16|  **42.42** |  **29.77**|**38.97**      |
> | mPLUG-Owl2| 10.33    |14.45| 39.56| 30.23  |38.18  |
> | VCD | 10.55 |15.61 |36.92|34.19 |38.80|
> | DoLA|10.33 |14.74|39.56| 30.47 |38.44 |
> |OPERA| 9.23 |  14.45| 39.78|28.37|38.26|
> |DAMO | **10.77**| **15.90**| **40.00**| **30.70** | **39.06**|
>
> # W3 & Q2 - More details about $\tau$: We have released the value of $\tau$ and conducted ablation experiment to evaluate the sensitivity of $\tau$
> *  $\tau$ is a hyperparameter that plays a key role in controlling the switch between coefficients $\beta_1$ and $\beta_2$. Through our experiments, we observed that the optimal value of $\tau$ varies across different models. We have provided our specific settings for $\tau$ in the next Table.
>     | Model | LLaVA1.5 |  INF-MLLM1   | mPLUG-Owl2 |
>     | -------- | --------- | --- | ---------- |
>     | $\tau$ value |-0.3| -0.6| -0.3 |
>
> * We evaluate the sensitivity of different $\tau$ on POPE (MSCOCO setting) dataset using LLaVA1.5 model and report F1 score. As shown in next Table, by setting $\tau$ to 0.3, 0, -0.3, and -0.6, we observe that the model's performance varies accordingly. This demonstrates the model's sensitivity to $\tau$. Notably, when $\tau$ is set to our default value of 0.3, the model achieves its best performance. You could check details in Appendix (E.1)
> | $\tau$      | 0.3  | 0   |   -0.3  | -0.6 |
> | ----------- | ---- | --- | --- | ---- |
> | Random      |   88.83 |  89.56  | **89.92** |  89.62         |
> | Popular     |  86.60|   86.43|  **87.07** |86.66 |
> | Adversarial |81.95|  81.56  |**82.07**| 81.78 |
>
>
>
>
>
>
>
> # Q3 - Further insights on the actual layers where the adaptive coefficient adjustment activates: We have presented more insights about the adjustments.
> * Figure 2(b) shows that our adaptive transition is effective in LLaVA 1.5, INF-MLLM1, and mPLUG-Owl2.
> * Due to time restraint, we analyzed 50 random samples from the POPE dataset using LLaVA 1.5 to study the transition behavior. Most data showed a switch from $\beta_1$ to $\beta_2$  between layers 23–24, reverting to $\beta_1$ around layers 28–29, with a few cases transitioning as early as layer 16. This supports our observation that hallucination issues often begin appearing in these layers.
>
> Hope these could address your concerns. Please let me know if you have any question.

---

> > ### Comment · Reviewer_5XME · 2024-11-22
> >
> > I maintain the initial rating for the following reasons:
> >
> > - The performance improvement on HallusionBench appears minor and lacks a substantial breakthrough.
> > - The model seems overly sensitive to hyperparameters, raising concerns that its performance may heavily depend on hyperparameter tuning.
> > - While the novelty is acknowledged, there is lingering uncertainty about whether the improvements are genuinely impactful or effective.

---

> ### Author Response · Authors · 2024-11-24
>
> Thanks for your reply.
>
> # Additional Question1: Limited performance improvement on HallusionBench
> *  As for HallusionBench, a really challenging benchmark, all these methods achieve a `limited performance achievements`, even worse than Regular Decoding. However, DAMO also achieves the almost greatest improvements  (except for fAcc on INF-MLLM1, where Regular Decoding achieves the best performance). Compared to results of other contrasting methods and OPERA, DAMO almost achieves the best performance. So we maintain that DAMO achieve great performance improvements on this challenging datasets. $\textbf {We hope you could take other methods' performance and the challenging nature of this benchmark into consideration}$.
>
> # Additional Question2:  The model seems overly sensitive to hyperparameters, raising concerns that its performance may heavily depend on hyperparameter tuning.
>
> Thank you for raising this concern. We would like to clarify that DAMO is not sensitive to hyperparameters. Here are the key points supporting this conclusion:
>
> * Ablation Experiments: We conducted ablation experiments involving $\tau$, $\beta_1$, and $\beta_2$, and the results, which can be found in Appendix E, demonstrate that DAMO's performance is stable across different hyperparameter settings. Specifically, both $\beta_1$ and $\tau$ show no sensitivity to variations in these hyperparameters. And maybe experiments related to $\beta_2$ made some confusion.
>
> * Clarification on $\beta_2$: We understand that the experiment involving $\beta_2$ might have caused some $\textbf{confusion}$. As shown in our original table, $\beta_2$ was tested at values of 0.10, 0.20, 0.40, and 0.80. While we observed a dramatic performance drop when $\beta_2$ was set too high, it’s important to note that this decline is $\textbf{not due to any issue with DAMO itself}$. When $\beta_2$ is set too high, it deviates from the initial goal of DAMO, which is to accumulate activations in a way that maintains visual semantics consistency. Excessive accumulation of activations causes the current layer's hidden states to become less relevant, thus $\textbf{interfering}$ with the model’s normal reasoning process. It should not be blamed to DAMO.
> * New Ablation Study on $\beta_2$: To more comprehensively evaluate the sensitivity of DAMO to $\beta_2$, we conducted a new ablation study where we tested $\beta_2$ values of 0.05, 0.10, 0.15, 0.20, 0.25, and 0.30. The results of this study are shown in the following Table. We believe this hyperparameter range provides a more balanced and fair evaluation of DAMO's sensitivity. Based on the results, it is clear that DAMO is stable and not sensitive to variations in $\beta_2$, reinforcing our earlier conclusions.
>
> | $\beta_2$ | 0.05 | 0.10    | 0.15    |  0.20   | 0.25    | 0.30 |
> | --------- | -------- | --- | --- | --- | --- | -------- |
> | MME score | 1509.79     |  1497.46   | 1496.43    |  1515.89   |   1496.83  | 1489.93     |
>
>
> # Additional Question3: While the novelty is acknowledged, there is lingering uncertainty about whether the improvements are genuinely impactful or effective:  Thank you for acknowledging the novelty of our approach. To address the lingering uncertainty about the impact and effectiveness of the improvements, we would like to highlight the following:
> * Quantitative Improvements: Our DAMO achieved a significant performance improvement in addressing hallucinations on various models across multiple benchmarks. This demonstrates that the improvements are both statistically and practically meaningful.
> * Unique Approach: Unlike traditional methods that rely on retraining or fine-tuning or contrasting decoding, our DAMO operates entirely during inference by directly modifying the hidden states. This not only solves the hallucination problem effectively but also introduces a lightweight, computationally efficient solution
> * Practical Relevance: The flexibility and adaptability of our method make it suitable for real-world applications where retraining is not feasible or contrasting decoding doesn't make sense(As for our experiment results, contrasting decoding seems achieve more limited improvements). Additionally, its ability to generalize across tasks and datasets highlights its robustness.
>
> Please let me know if you have any question.

---

> ### Author Response · Authors · 2024-11-26
> **A kindly reminder**
>
> Dear Reviewer 5XME,
>
> We really appreciate your valuable advice and questions.
>
> And we are really excited to hear that our novelty could be acknowledged. Meanwhile, we also notice your new questions and concerns. And we have refined our response to further clarify the $\textbf{confusion and misclarification}$. We hope you could $\textbf{take a look}$ at our new response and  $\textbf{give some more valuable advice}$ if possible.
>
> Many thanks,
>
> Authors of Submission11551

---

> > ### Comment · Reviewer_5XME · 2024-11-28
> >
> > ### Key Points and Required Clarifications:
> > #### 1. Sensitivity to \( \tau \):
> > - Basis for Claiming \( \tau \) is Not Sensitive:
> >   - The authors argue that \( \tau \) is not sensitive, but when \( \tau = 0.30 \), the Random F1 Score (88.83) is lower than at other decoding values. How does this support the claim of insensitivity?
> >   - Why was the evaluation of \( \tau \)’s sensitivity conducted using only the POPE dataset? Extending this analysis to other datasets could provide a stronger foundation for the claim.
> > #### 2. Sensitivity to \( \beta_1 \):
> > - Conflict Between Sensitivity and MME Benchmark Results:
> >   - Appendix Table 10 shows that when \( \beta_1 = 0.20 \), the Total Score on the MME benchmark (1462) is lower than other decoding methods. How does this align with the claim that \( \beta_1 \) is not sensitive?
> > #### 3. Lack of Trends for \( \beta_2 \):
> > - Missing Explanation of \( \beta_2 \)’s Non-linear Relationship:
> >   - The authors assert that excessively high values of \( \beta_2 \) should be avoided, and the new ablation study is appreciated. However, the additional results show no clear trends regarding the impact of \( \beta_2 \) on performance. Is the relationship between \( \beta_2 \) and performance inherently non-linear or inconsistent? Clarifying this is essential to understanding the role of \( \beta_2 \).
> > ---
> > These points highlight critical gaps in the rebuttal, where additional evidence or explanations are necessary to justify the authors’ claims of insensitivity to hyperparameters.

---

> ### Author Response · Authors · 2024-12-02
> **Further response to Reviewer 5XME (1/2)**
>
> # Q1: Sensitivity to $\tau$:
> > The authors argue that ( \tau ) is not sensitive, but when ( \tau = 0.30 ), the Random F1 Score (88.83) is lower than at other decoding values. How does this support the claim of insensitivity?
>
> - Thank you for your question, and we want to clarify it more. The sensitivity experiment was designed to evaluate `how DAMO's performance varies with different hyperparameters`. Our results demonstrate that DAMO is not highly sensitive to the choice of $\tau$. As shown in Appendix (E.1), the results on the POPE dataset indicate that changes in $\tau$ have little effect on DAMO's performance. Specifically, when $\tau$ is varied across values of 0.3, 0, -0.3, and -0.6, the F1 score remains relatively stable, with no dramatic fluctuations. In the Random, Popular, and Adversarial settings, the maximum changes in F1 score are 1.09, 0.64, and 0.51, respectively. This suggests that DAMO is not highly sensitive to $\tau$. Additionally, our experiments show that the default value of $\tau$ provides the optimal performance.
>
> > Why was the evaluation of ( \tau )’s sensitivity conducted using only the POPE dataset? Extending this analysis to other datasets could provide a stronger foundation for the claim.
>
> - Thank you for your question. Due to time and resource constraints, we initially conducted some ablation experiments using a single dataset. However, we are happy to provide additional results. Given the high cost of using the GPT-API to evaluate HallusionBench, we have included more ablation experiments using the MME dataset on the LLaVA1.5 model. In these experiments, similar to our previous setup, we varied $\tau$ across the values 0.3, 0, -0.3, and -0.6 to assess how DAMO's performance is sensitive to changes in $\tau$. As shown in the next Table, DAMO also shows no sensitivity to $\tau$ on MME dataset.
>
> | $\tau$    | 0.3  | 0    | -0.3    | -0.6 |
> | --------- | ---- | --- | --- | -------- |
> | MME Score | 1480.73 |  1491.14   |    1515.89 |    1475.34  |
>
> # Q2: Sensitivity to ( $\beta_1$ ):
> > Appendix Table 10 shows that when ( \beta_1 = 0.20 ), the Total Score on the MME benchmark (1462) is lower than other decoding methods. How does this align with the claim that ( \beta_1 ) is not sensitive?
>
> - Thanks for your question. Similar to previous confusion, `our sensitivity experiment was also designed to evaluate how DAMO's performance varies with different hyperparameters`. As shown in Table 10, DAMO's performance doesn't have dramatic change with the variation of $\beta_1$.
> - Additionally, we also add two group experiment to further explore DAMO's sensivity to $\beta_1$. We also provide experiments using INF-MLLM1 and mPLUG-Owl2 models to demonstrate that DAMO is not sensitive to $\beta_1$. The first table shows the results on INF-MLLM1 model using MME dataset. And the second table shows the results on mPLUG-Owl2 using the same dataset. We believe the additional results demonstrates that DAMO's performance is not sensitive to the $\beta_1$.
>
> | $\beta_1$ | 0.05 | 0.10 | 0.15 | 0.20 | 0.25 | 0.30 | 0.35 | 0.40 |
> | --------- | ---- | ---- | ---- | ---- | ---- | ---- | ---- | ---- |
> | MME Score | 1498.25 | 1506.87 |  1515.15    |     **1520.74** |  1514.70    |    1504.36  |   1507.11   |   1515.61   |
>
>
> | $\beta_1$ | 0.50    | 0.55    | 0.60        | 0.65    | 0.70    | 0.75    |
> | --------- | ------- | ------- | ----------- | ------- | ------- | ------- |
> | MME score | 1419.36 | 1419.62 | **1437.46** | 1433.69 | 1435.30 | 1432.52 |

---

> > ### Author Response · Authors · 2024-12-02
> > **Further response to Reviewer 5XME (2/2)**
> >
> > # Q3: Lack of Trends for ( \beta_2 ):
> > > The authors assert that excessively high values of ( \beta_2 ) should be avoided, and the new ablation study is appreciated. However, the additional results show no clear trends regarding the impact of ( \beta_2 ) on performance. Is the relationship between ( \beta_2 ) and performance inherently non-linear or inconsistent? Clarifying this is essential to understanding the role of ( \beta_2 ).
> >
> > - Thank you for your question. We agree that this is an interesting point, and we'd like to clarify it here.
> > - Firstly, the relationship between $\beta_2$ and performance cannot be simply characterized as non-linear. The reason we introduced the gating mechanism between $\beta_1$ and $\beta_2$ is to prevent the accumulation of incorrect information through momentum. As described in the methods section, when the direction of the information learned at the current step aligns with the accumulated direction, we focus on the current layer's learning. This ensures that the inference process remains unaffected and minimizes the risk of hallucinations.
> > - However, when we detect hallucinations, we switch to $\beta_2$, which places more emphasis on the information accumulated through momentum, reducing the proportion of hallucinated information from the current layer. The setting of $\beta_2$ is not intended to follow a simple linear or non-linear relationship with performance. Instead, our experiments provide practical examples of $\beta_2$ values that can serve as prior knowledge for future applications and model transfers.
> >
> > Hope these explanations could address your concerns.

---

### Official Review · Reviewer_au8F · 2024-11-03

**Soundness:** 3
**Presentation:** 2
**Contribution:** 2
**Rating:** 5
**Confidence:** 3

**Summary:**

This paper observes that VLMs tend to make prediction shifts in the last few layers, which leads to a surge in hallucinative outputs. The authors propose a decoding strategy inspired by the momentum analogy used in gradient descent-based optimizers, which enforces decoding consistency adaptively across layers during forward passes. The proposed method outperforms existing approaches on several public datasets.

**Strengths:**

1. The motivation of hallucinations frequently emerge in the later layers seems interesting.
2. The main technical pipeline is clear.

**Weaknesses:**

1. Writing and diagrams require improvement.
1). Large Vision-Language Models (VLMs) mentioned in the abstract abbreviated as LVLMs would be better.
2). A single-paragraph abstract would improve conciseness.
2). Figure 1(c) requires a more detailed description.
3). In Section 2, lines 151–152, the statement “However, these methods do not correct hallucinations during the inference process” could be reconsidered. In my view, methods like VCD, which contrast output distributions, are indeed part of the inference process.
4). It would improve readability if the introduction were revised to emphasize the primary contributions of this work more clearly.

2. Could you increase the number of instances shown in Figure 1(b) to provide a more comprehensive view?

3. Section 3.2, line 198-200, "VLMs are already proficient in capturing detailed visual information, so further intensifying image-text fusion is unnecessary." how to validate "VLMs are already proficient in capturing detailed visual information" from your results?

4. Section 3.4, line 261-261, "Varying the starting layer for refinement can enhance different model capabilities (e.g., layer 16 excels in OCR tasks, while layer 24 improves positional perception)." (e.g., position scores do not consistently increase, nor do OCR scores consistently decrease). This makes it challenging to conclude that varying the starting layer directly enhances distinct capabilities. Could you provide additional evidence to support this claim?

5. The performance improvement appears modest. For instance, on the MME benchmark, the score of 1515.89 is slightly lower than OPERA’s 1518.36 on the LLaVA baseline. Additionally, it would strengthen the evaluation if more hallucination benchmarks, such as CHAIR, were included.

6. I am concerned about the quality of text generated at preceding layers. Further evaluation metrics for text quality, such as BLEU or other relevant scores, would provide a clearer understanding.

**Questions:**

My primary concern lies in the performance and  the quality of text generated at preceding layers. I will be happy to raise my score if my current questions and concerns can be addressed.

---

> ### Author Response · Authors · 2024-11-20
> **Thank you! (1/2)**
>
> We sincerely appreciate  your review and the valuable suggestions provided.
>
> # W1  -  Issues about writing and diagrams:  Thanks for your valuable advice and we have revised in the updated version
>
> * 1) We have revised this abbreviation from VLMs to LVLMs. You could check in the updated version.
> * 2) We have revised our abstract in a single paragraph. For figure 1(c), we have abstained this image and detailed information related to it is shown in our motivation part.
> * 3) Sorry for the confusion. We will clarify this statement here and revise it in the camera-ready version. Specifically, what we aim to illustrate is that while both DoLA and VCD predict the next token based on contrasting output distributions, neither modifies the hidden states during inference. In contrast, our DAMO focuses on addressing hallucinations by directly refining the hidden states throughout the inference process. And we think that is the fundamental difference. We hope this explanation addresses your question.
> * 4) We have summarized our primary contributions in the updated version. And we show them here: a) We identify that hallucinations in Large Vision-Language Models (LVLMs) primarily occur during the inference process and are driven by localized surges in the later layers, which suppress visual information.  b) We introduce Decoding by Accumulating Activations Momentum (DAMO), a novel approach that reduces hallucinations by accumulating activation momentum, significantly
> improving visual grounding in LVLMs across multiple benchmarks.  Thanks again for your advice.
>
> # W2  -  More instances related to Figure 1(b):  We have presented 2 more instances in Appendix (C).
> * Both instances exhibit the same issue as Figure 1b. In regular decoding, the probability of the correct token during the inference process first increases and then decreases across layers, resulting in the loss of visual information. In contrast, our DAMO achieves visual semantics consistency, where the probability of the correct token shows an upward trend.
>
> # W3 - Validation about certain statement: We have validated this statement from qualitative analysis and quantitative results.
> * Qualitative analysis: As shown in Figure 1 (b), the probility of token `two` gets the highest at 26-th layer, it drops dramatically in the following layers. In other words, the correct visual information can only noticeably occur during the inference process if the vision encoder has proficiently captured the visual information. The 2 more instances we presented could also support this claim.
> * Quantitative results: Our proposed DAMO doesn't finetune any parameters of visual enconder and languane model, instead of accumulating the language model activations across layers during inference. So it could not get any performance enhancement if the visual encoder didn't capture sufficient image information.
>
> # W4 - Evidence to support that varying the starting layer directly enhances distinct capabilities: We present two results to support this claim.
> * As shown in Table 2), the improvement from 1508.39 (with a fixed starting layer at 24, as shown in Table 1) to 1515.89 (using our adaptive activation refinement mechanism, demonstrates that our method adaptively selects the optimal starting layer for different tasks, outperforming any fixed starting layer approach.
> * Figure 2(a) also demonstrates the effect of our adaptive activation refinement mechanism. Our adaptive starting layer selection outperfoms all other fixed starting layers setting.
> * We further evaluate our proposed adaptive activation refinement mechanism.  In addition to testing the effectiveness of our adaptive activation refinement mechanism on LLaVA1.5, we further evaluate it on INF-MLLM1 by setting the starting layer to the 16th, 20th, 24th, and 28th layers. As shown in the next Table,which demonstrates the effectiveness of our method in selecting the starting layer adaptively based on different question scenarios. Furthermore, we believe that finding a method to achieve optimal performance for every individual task even question is inherently challenging. Therefore, we use the overall performance as a metric to evaluate the effectiveness of our adaptive activation refinement mechanism. You could check details in Appendix (E.4).
> | Starting Layer | 16-th |   20-th  |  24-th   |  28-th   | Adaptive |
> | -------- | -------- | --- | --- | --- | -------- |
> | MME score     | 1509.92     |  1516.65   |  1513.56   |  1507.53   | **1520.74** |

---

> ### Author Response · Authors · 2024-11-20
> **Thank you! (2/2)**
>
> # W5 - Modest performance improvement and hallucination-targeted benchmark: We will further compare our results and conduct more experiments.
> * On the MME benchmark, it is true that DAMO's score is slightly lower than OPERA's score when using the LLaVA model. However, DAMO outperforms all other methods, including OPERA, on both the INF-MLLM1 and mPLUG-Owl2 models across the MME and POPE datasets, leading to a significant overall performance improvement. Additionally, due to the beam search setting, OPERA has a much higher memory consumption and inference latency compared to DAMO during the inference process.
> * Due to the inconsistent experimental settings in previous works using CHAIR (e.g., some studies randomly selected images for comparison), we have used a new hallucination benchmark called HallusionBench for evaluation. HallusionBench offers a well-established evaluation framework that facilitates consistent and reliable experimentation. We present results here and we believe this results could demonstrate our DAMO's effectiveness on hallucination-targeted benchmarks. Details can be found in Appendix (H).
> | Model     | qAcc   | fAcc    | easy aAcc | hard aAcc | aAcc    |
> | --------- | ------ | ------- | --------- | --------- | ------- |
> | LLaVA1.5  | 8.13 | 14.16 | 36.92   | 27.91    | 35.43 |
> | VCD       | 9.01  | 14.16 | 36.70   | 29.53   | 35.96 |
> | DoLA      | 9.45 | 14.45 | 36.70   | 29.07   | 36.23 |
> | OPERA     | 8.57 | 14.45 | 37.14   | 29.07   | 35.96  |
> | DAMO      |   **9.67**     | **14.74** |**37.14**|**29.77**     |  **36.58**       |
> | INF-MLLM1 | 7.69 | **14.45** | 41.98    | 29.07   | 38.62 |
> | VCD       | 7.25 | 13.01 | 40.66  | 28.37   | 37.73 |
> | DoLA      | 7.03 |  13.01 | 40.22  | 29.30|38.00 |
> | OPERA     | 7.25|  13.87| 41.76| 27.44 | 37.91 |
> | DAMO      |     **8.13** |  14.16|  **42.42** |  **29.77**|**38.97**      |
> | mPLUG-Owl2| 10.33    |14.45| 39.56| 30.23  |38.18  |
> | VCD | 10.55 |15.61 |36.92|34.19 |38.80|
> | DoLA|10.33 |14.74|39.56| 30.47 |38.44 |
> |OPERA| 9.23 |  14.45| 39.78|28.37|38.26|
> |DAMO | **10.77**| **15.90**| **40.00**| **30.70** | **39.06**|
>
> # W6 & Q1 - Quality of text generated at preceding layers: We present an example to show the token variations across layers during inference and use GPT-API to evaluate the text quality.
> * Sorry for the confusion, but our method differs from the early exit approach. While early exit stops inference at intermediate layers to predict the next token, which may lead to lower text quality due to insufficient inference, our DAMO accumulates hidden state activations in a momentum-like manner until reaching the final layer, where it predicts the next token. Therefore, theoretically, the text quality will not degrade.
> * To address your concern about text quality, we provide an example extending Figure 1, where we decode each word of the entire sentence predicted at each layer. As expected, the model consistently generates higher-quality text at the later layers, aligning with common understanding. Moreover, DAMO addresses hallucinations during the inference process. Different from early exit technique, both Regular Decoding and DAMO use the logits generated from last layer to predict next token, so it is better to focus on the probability variations across layers. You can check the details in Appendix (F.1).
>
> *  We are also happy to evaluate the text quality generated by DAMO. For convenience, we used the GPT-API to evaluate the quality of the generated text from several perspectives: Fluency, Coherence, Grammar and Syntax, and Vocabulary Usage. Detailed experimental settings can be found in the Appendix (F.2). The results demonstrate that the text generated by DAMO is of high quality.
> |   Method  |  Regular   |  VCD   | DoLA | OPERA | DAMO |
> | --- | --- | --- | -------- | -------- | -------- |
> |  Score   |   7.9  | 7.8  | 8.1  | 8.0 | 8.1   |
>
> I hope this could address your concerns. Please let me know if you have any question related to text quality evaluation.

---

> > ### Comment · Reviewer_au8F · 2024-11-23
> >
> > Thank you for the detailed response. I reviewed the revised paper and the additional comments. While some of my concerns were addressed, the performance improvements remain limited, and the method shows significant sensitivity to hyperparameters. This makes it hard to demonstrate the effectiveness of the approach. Therefore, I will maintain my initial rating.

---

> ### Author Response · Authors · 2024-11-24
> **Further response to Reviewer au8F**
>
> Thanks for your reply! And we would like to clarify more about your additonal questions.
>
> # Additional Question1: Question about limited performance improvements: We believe that our DAMO achieves a great performance improvement. The reasons could be summarized as follows:
> * Our DAMO performs excellently on $\textbf{POPE}$ dataset. Taking mPLUG-Owl2 model as an example, under A-OKVQA dataset and ramdom setting, DAMO's F1 score is `86.03`, while VCD is `84.16`, DoLA is `84.07`, showcasing remarkable performance improvement in addressing object hallucinations. And DAMO also performs excellently on other models, like  LLaVA 1.5, and INF-MLLM1. What's more, as shown in Table 3, VCD, DOLA and OPERA sometimes show $\textbf{performance decreasement}$ compared to Regular Decoding. However, DAMO outperforms Regular Decoding consistently across three models and two datasets.  What need to mention is that `temperature` is set to `0` for all experiments, leading to convincing and fair comparisons. So for all these experimental results, we believe that DAMO achieves $\textbf{great performance improvement}$.
>
> * Meanwhile, DAMO also performs well on $\textbf{MME}$ dataset, proving our DAMO's ability in generalized task and hallucinations task. As shown in Table 2, DAMO achieves the best perforances across three models except for LLaVA1.5. However, on LLaVA1.5, with only 2.47 scores improvement, OPERA introduces more 7.9 GB memory cost and more inference time cost (which is really time-cost compared to all methods because it adopts `beam_search` to address hallucinations) compared to DAMO. And on INF-MLLM1, DAMO even outperforms OPERA with 27.59 scores improvements. So for this experimental results, DAMO is cost-efficient and effective. So we also believe that DAMO does achieve great performance improvements.
>
> * As for $\textbf{HallusionBench}$, a really challenging benchmark, all these methods achieve a `limited performance achievements`, even worse than Regular Decoding. However, DAMO also achieves the almost greatest improvements  (except for fAcc on INF-MLLM1, where Regular Decoding achieves the best performance). Compared to results of other contrasting methods and OPERA, DAMO almost achieves the best performance. So we maintain that DAMO achieve great performance improvements on this challenging datasets. $\textbf {We hope you could take other methods' performance and the challenging nature of this benchmark into consideration}$
>
> # Additional Question2: Method shows sensitivity to hyperparameters: We have clarified some key points.
> Thank you for raising this concern. We would like to clarify that DAMO is not sensitive to hyperparameters. Here are the key points supporting this conclusion:
>
> * Ablation Experiments: We conducted ablation experiments involving $\tau$, $\beta_1$, and $\beta_2$, and the results, which can be found in Appendix E, demonstrate that DAMO's performance is stable across different hyperparameter settings. Specifically, both $\beta_1$ and $\tau$ show no sensitivity to variations in these hyperparameters. And maybe experiments related to $\beta_2$ made some confusion.
>
> * Clarification on $\beta_2$: We understand that the experiment involving $\beta_2$ might have caused some $\textbf{confusion}$. As shown in our original table, $\beta_2$ was tested at values of 0.10, 0.20, 0.40, and 0.80. While we observed a dramatic performance drop when $\beta_2$ was set too high, it’s important to note that this decline is $\textbf{not due to any issue with DAMO itself}$. When $\beta_2$ is set too high, it deviates from the initial goal of DAMO, which is to accumulate activations in a way that maintains visual semantics consistency. Excessive accumulation of activations causes the current layer's hidden states to become less relevant, thus $\textbf{interfering}$ with the model’s normal reasoning process. It should not be blamed to DAMO.
> * New Ablation Study on $\beta_2$: To more comprehensively evaluate the sensitivity of DAMO to $\beta_2$, we conducted a new ablation study where we tested $\beta_2$ values of 0.05, 0.10, 0.15, 0.20, 0.25, and 0.30. The results of this study are shown in the following Table. We believe this hyperparameter range provides a more balanced and fair evaluation of DAMO's sensitivity. Based on the results, it is clear that DAMO is stable and not sensitive to variations in $\beta_2$, reinforcing our earlier conclusions.
>
> | $\beta_2$ | 0.05 | 0.10    | 0.15    |  0.20   | 0.25    | 0.30 |
> | --------- | -------- | --- | --- | --- | --- | -------- |
> | MME score | 1509.79     |  1497.46   | 1496.43    |  1515.89   |   1496.83  | 1489.93     |
>
> We hope these explanations could address your concerns. Please let us know if you have any question.

---

> > ### Comment · Reviewer_au8F · 2024-11-26
> >
> > Thank you for addressing my previous comments. I appreciate the additional responses and analyses provided. However, I have three concerns:
> > 1. The performance of DAMO appears inconsistent across different datasets. For instance, in the MME hallucination subtasks—Existence, Count, Position, and Color—DAMO significantly underperforms compared to OPERA.
> >
> > 2. On the MME dataset using the LLaVA1.5 model, the performance is sensitive to the choice of $\beta_1$ . For example, a change from 0.05 to 0.2 results in a significant performance difference, reaching up to 111.94.
> >
> > 3. The new ablation experiments for $\beta_2$ seem problematic. For mPLUG-Owl2 M and mPLUG-Owl2 P,  $\beta_2$ is set to 0.8 and 0.6, respectively. These values contradict the stated new hyperparameter range and your observation that "a dramatic performance drop occurs when $\beta_2$ is set too high."

---

> > > ### Author Response · Authors · 2024-12-02
> > > **Further response to Reviewer au8F**
> > >
> > > # Q1: The performance of DAMO appears inconsistent across different datasets. For instance, in the MME hallucination subtasks—Existence, Count, Position, and Color—DAMO significantly underperforms compared to OPERA.
> > >
> > >
> > > Thank you for your question.
> > >
> > > - Regarding the MME hallucination subtasks you mentioned—Existence, Count, Position, and Color—DAMO does not significantly underperform compared to OPERA. As shown in Table 2, DAMO actually achieves the best score in the Existence subtask across all three models. For the Count subtask, all methods, including OPERA, perform worse than regular decoding. In the mPLUG-Owl2, DAMO also achieves the best score, similar to OPERA. In the Position subtask, DAMO outperforms OPERA by a significant margin in both INF-MLLM1 and mPLUG-Owl2. While OPERA does outperform DAMO in a few tasks, overall, DAMO leads in performance across all three models.
> > >
> > > * Additionally, as shown in Table 2, DAMO demonstrates a significant advantage in memory usage, with OPERA requiring much more memory. Furthermore, since OPERA uses beam search to handle hallucination, it also requires significantly more inference time compared to DAMO (we have measured inference times in our experiments). We believe that considering efficiency, alongside performance, provides a fairer comparison.
> > >
> > > # Q2: On the MME dataset using the LLaVA1.5 model, the performance is sensitive to the choice of $\beta_1$.  For example, a change from 0.05 to 0.2 results in a significant performance difference, reaching up to 111.94.
> > >
> > > Thank you for your concerns.
> > > * We believe that the observed sensitivity to the choice of $\beta_1$ is not specific to the DAMO method, but rather reflects the inherent sensitivity of the LLaVA model to modifications of the hidden states.
> > > * To further clarify this, we will also provide experiments using INF-MLLM1 and mPLUG-Owl2 models to demonstrate how these models also exhibit different sensitivity to the choice of $\beta_1$ from LLaVA1.5.
> > >
> > >
> > >
> > > | $\beta_1$ | 0.05 | 0.10 | 0.15 | 0.20 | 0.25 | 0.30 | 0.35 | 0.40 |
> > > | --------- | ---- | ---- | ---- | ---- | ---- | ---- | ---- | ---- |
> > > | MME Score | 1498.25 | 1506.87 |  1515.15    |     **1520.74** |  1514.70    |    1504.36  |   1507.11   |   1515.61   |
> > >
> > >
> > > | $\beta_1$ | 0.50    | 0.55    | 0.60        | 0.65    | 0.70    | 0.75    |
> > > | --------- | ------- | ------- | ----------- | ------- | ------- | ------- |
> > > | MME score | 1419.36 | 1419.62 | **1437.46** | 1433.69 | 1435.30 | 1432.52 |
> > >
> > >
> > > - The first table presents the results on the INF-MLLM1 model using the MME dataset, while the second table shows the results on mPLUG-Owl2 with the same dataset. In the first table, the maximum performance change is 22.49 (when $\beta_1$ is set to 0.05 and 0.20), and in the second table, the maximum performance change is 18.1 (when $\beta_1$ is set to 0.50 and 0.60). We believe that these new results could demonstrate that DAMO is not sensitive to $\beta_1$.
> > >
> > > # Q3: The new ablation experiments for $\beta_2$ seem problematic. For mPLUG-Owl2 M and mPLUG-Owl2 P, $\beta_2$ is set to 0.8 and 0.6, respectively. These values contradict the stated new hyperparameter range and your observation that "a dramatic performance drop occurs when $\beta_2$ is set too high."
> > >
> > > Thank you for pointing this out. We apologize for any confusion caused. Our previous statement regarding a "dramatic performance drop when $\beta_2$ is set too high" was specifically focused on the LLaVA1.5 model, where we observed such behavior. As we illustrated in your last concern, this behavior lies in the inherent sensitivity of LLaVA model instead of DAMO method.  However, for INF-MLLM1 and mPLUG-Owl2, this issue does not occur, and the values of $\beta_2$ used in these models (0.8 for mPLUG-Owl2 M and 0.6 for mPLUG-Owl2 P) are within the expected range for their respective architectures.
> > >
> > > Hope these explanation could address your concern.

---

> ### Author Response · Authors · 2024-11-26
> **A kindly reminder**
>
> Dear Reviewer au8F,
>
> We really appreciate your valuable advice and questions.
>
> As you mentioned, our first response has addressed your primary concern about the text quality and other quesitons. And we are really excited to hear that. Meanwhile, we also notice your new questions and concerns. And we have refined our response to further clarify the $\textbf{confusion and misclarification}$. We hope you could $\textbf{take a look}$ at our new response and  $\textbf{give some more valuable advice}$ if possible.
>
> Many thanks,
>
> Authors of Submission11551

---

### Meta-Review · Area_Chair_dMu2 · 2024-12-20

**Metareview:**

This work deals with the issue of hallucinations in Large Vision-Language Models (VLMs) by analysing their layer-wise prediction tendencies and decoding schemes. The authors designed a decoding strategy ensuring adaptive decoding consistency over layers. Overall, the method is good and interesting. The paper provides an interesting observation that VLMs can often generate accurate predictions in the initial layers, but suffers hallucinations in later layers. Though there are some weaknesses in the paper, which could be addressed in the camera ready version.

**Additional Comments On Reviewer Discussion:**

Authors provided detailed response addressing reviewers concerns, including adding detailed analysis and comparisons. Most of the concerns have been well replied.

---

### Decision · Program_Chairs · 2025-01-22

Accept (Poster)